# Improving the precipitation accumulation analysis using lightning measurements and different integration periods

E. Gregow[1], A. Pessi[2], A. Mäkelä[1] and E. Saltikoff[1]

[1]Finnish Meteorological Institute, P.O. Box 503, FIN-00101 Helsinki, Finland

[2]Vaisala, 3 Lan Dr., Westford, MA 01886, USA

*Correspondence to:* E. Gregow (erik.gregow@fmi.fi)

**Abstract.** The focus of this article is to improve the precipitation accumulation analysis, with special focus on the intense precipitation events. Two main objectives are addressed: (i) the assimilation of lightning observations together with radar and gauge measurements and (ii) the analysis of the impact of different integration periods in the radar-gauge correction method. The article is a continuation of previous work in the same research-field, by Gregow et al. (2013).

A new lightning data assimilation method has been implemented and validated within the finnish meteorological institute - local analysis and prediction system. Lightning data do improve the analysis when no radars are available, and even with radar data, lightning data have a positive impact on the results.

The radar-gauge assimilation method is highly dependent on statistical relationships between radar and gauges, when performing the correction to precipitation accumulation field. Here we investigate the usage of different time integration intervals: 1, 6, 12, 24 hours and 7 days. This will change the amount of data used and affect the statistical calculation of the radar-gauge relations. Verification shows that the real-time analysis using the 1 hour integration time length gives the best results.

1 Introduction

Accurate estimates of accumulated precipitation are needed for several applications, such as flood protection, hydropower, road- and fire-weather models. In Finland, one of the economically most relevant users of precipitation is hydropower industry. Between 10 and 20% of Finnish annual electric power production comes from hydropower, depending on the amount of precipitation and water levels in dams and water reservoirs. In order to maintain correct calculation of the energy supplied to customers and to avoid (or at least minimize) the environmental risks and economical losses during extreme precipitation and flooding events, a profound analysis of the expected water amounts in dams and reservoirs from catchment-areas is needed. The current hydropower strategy of Finland is to increase capacity by improving the efficiency of existing plants through technical adjustments. The maintenance and planning of proper dam structures need the most up-to-date information about the rain rates to be able to adjust the regulation functions of the dams, both for the current and the changing climatic conditions (IPCC-AR5, 2013).

Often, the accumulated precipitation values are based on pure radar analysis, unless there exists a surface gauge observation in the immediate surroundings. Radar echoes are related to rainfall rate and thereafter transformed into accumulation values. However, such conversions are based on general empirical relations, which are not suitable for all meteorological cases (e.g. depending on precipitation type; Koistinen and Michelson, 2002).

Radar reflectivity can in some cases suffer from poor quality, resulting from electronic mis-calibration, beam blocking, clutter, attenuation and overhanging precipitation (Saltikoff et al., 2010), which results in poor estimations of the precipation accumulation. In some cases the radar can even be missing, e.g. during maintenance, upgrading or due to technical problems. Especially during thunderstorms there is a potential of radar disturbances, either in form of missing data due to interruptions in electricity and telecommunication systems, or in form of quality issues such as attenuation, due to intervening heavy precipitation.

The research of combining radar and surface observations, to perform corrections to precipitation accumulation, is well explored. Many have made developments in this field and much literature is available, for example Sideris et al. (2014), Schiemann et al. (2011) and Goudenhoofdt and Delobbe (2009). In general, combining radar and rain gauge data is very difficult in the vicinity of heavy, local rain cells (Einfalt et al., 2005). Recently, Jewell and Gaussiat (2015) compared performances of different merging schemas, and noted a large difference between convective and stratiform situations. In their study, the non-parametric kriging with external drift outperformed other methods in accumulation period of 60 minutes. Wang et al (2015) developed a sophisticated method for urban hydrology, which preserves the non-normal charactersitics of the precipitation field. They also noticed that common methods have a tendency to smooth out the important but spatially limited extremes of precipitation.

Comparing radars and gauges, an additional challenge arises from the different sampling sizes of the instruments. Radar measurement volume can be several kilometers wide and thick (one degree beam is approximately 5 kilometres wide at 250 kilometres), while the measurement area of a gauge is 400 cm$^2$ (weighing gauges) or 100 cm$^3$ (optical instruments). Part of the disparateness of radar and gauge measurements is due to variability of the raindrop size distribution within area of a single radar pixel. Jaffrain and Berne (2012) have observed variability up to 15% of the rainrate in a 1x1 km pixel, with timesteps of 1 minute.

Lightning is associated with convective precipitation, but in areas where a large portion of precipitation is stratiform, lightning data alone are not adequate for precipitation estimation. Although convective events contribute only a fraction of the annual precipitation amount, they might be important during flooding events. However, lightning has been used to complement and improve other datasets. Morales and Agnastou (2003) combined lightning with satellite-based measurements to distinguish between convective and stratiform precipitation area and achieved a remarkable 31% bias reduction, compared to satellite-only techniques. Lightning has also been assimilated to numerical weather prediction (NWP) models, using nudging techniques, or improving the initialization process of the model. This can be done by blending them with other remote sensing data to create heating profiles (e.g. estimating the latent heat release when precipitation is condensed). Papadopulos et al. (2005) used lightning data to identify convective areas and then modified the model humidity profiles, allowing the model to produce convection and release latent heat using its own convective parameterization scheme. They combined lightning with 6-hourly gauge data, within a mesoscale model in the Mediterranean area, and showed improvement in forecasts up to 12 hours lead time. Pessi and Businger (2009) derived a lightning-convective rainfall relationship over the North Pacific Ocean and used it for latent heat nudging method in an NWP model. They were able to improve the pressure forecast of a North Pacific winter storm significantly.

Our situation is different from the above mentioned experiments because lightning activity is usually low in Finland, compared to warmer climates (Mäkelä et al., 2011). Also, our analysis area already has a good radar coverage and relatively evenly distributed network of 1 hour gauge measurements. However, if we want to enlarge the analysis area, we will soon go to either sea areas or neighbouring countries where availability of radar data and frequent gauge measurements is low. We also anticipate the usefulness of lightning data as backup

plan in the occasions when radar data is either missing or of detoriated quality. Even though these occasions are rare, they often occur on days when detailed precipitation estimates are of great interest. Thunderstorms producing heavy localized rainfall are also often producing heavy winds, causing unavailability of radar data due to breaks on electricity and data communications. Our principal goal is to have as good analysis as possible, which is different from having a best analysis to start a model.

Gregow et al. (2013) have demonstrated the benefit of assimilating different data sources (radars and gauges) in precipitation estimation. The largest uncertainties were observed during heavy convective rainfall. These are the situations when lightnings occur. The accumulation process is based on radar reflectivity field, where gauges corrects the initial field, e.g. if there is no reflectivity field there is no accumulation (gauges are not used alone). To improve the spatially accurate real-time precipitation analysis new methods are adopted by fusion of weather

radar, lightning observations and rain gauge information in novel ways. This leads to better possibilities in estimating convective rainfall events (i.e. > 5 mm/h) and the accumulated precipitation, for the benefit of hydropower management and other related application areas. The work reported here has been performed using the Local Analysis and Prediction System (LAPS), which is used operationally in the weather service of Finnish Meteorological Institute (FMI). Testing new approaches in an operational system has its challenges. For

example, it is not possible to exclude a large amount of independent reference stations. Also the possibilities to rerun cases with different settings have been limited. The major benefit of working in an operational environment is that we can be sure that we only use data and methods which are operationally available and feasible.

In this article the observational datasets are described in chapter 2. New methods on how to calculate the

precipitation accumulation is handled in chapter 3, and the results and discussion are shown in chapters 4 and 5, respectively.

**2 Observations and instrumentation**

Here we describe the three data sources employed in this study: rain gauge-, radar- and lightning observations and the verification periods used in this study.

**2.1 Rain gauge observations**

Rain gauges provide point observations of the accumulation. They are usually considered more accurate than radar, as point values, and are frequently used to correct the radar field (Wilson and Brandes, 1979). The surface precipitation network (in total 472 stations) consists of standard weighting gauges and optical sensors mounted on road-weather masts. Since 2015, FMI manages 102 stations instrumented with the weighting gauge OTT

Messtechnik Pluvio2. The Finnish Transport Agency (FTA) runs 370 road-weather stations with optical sensor

measurements (Vaisala Present Weather Detectors models PWD22 and, to some extent, PWD11). The precipitation intensity is measured in different time intervals which are summed up to 1 hour precipitation accumulation information. Uncertainties and more detailed information can be found in Gregow et al. (2013). If measurements consistently indicate poor data quality, either manually identified from station error-logs or by inspecting the data, those stations are blacklisted within the LAPS process and do not contribute to the precipitation accumulation analysis. Hereafter in this article, the weighting gauges and road-weather measurements are indistinctly called gauges and their placement in Finland is shown in Fig. 1a.

## 2.2 The radar data

As of summer 2016, FMI operated ten C-band Doppler radars (newest one operational since late 2015). All but one station (VIM in western Finland; see Fig. 1b) are dual-polarization radars. At the moment, the quantitative precipitation estimation based on dual-polarization is not used operationally in FMI, but the polarimetric properties contribute to the improved clutter cancellation (i.e. removal of non-meteorological echoes, especially sea clutter, birds and insects). In southern Finland, the distance between radars is 140–200 km, but in the north, the distance between stations LUO and UTA is 260 km. The location of the radars and the coverage is shown in Fig. 1b. As Finland has no high mountains, the horizon of all the radars is near zero elevation with no major beam blockage, and, in general, the radar coverage is very good except in the most northern part of the country. The Finnish radar network does have a very high system utilization rate (e.g. no interruption). During year 2014 and 2015 the utilization rate was > 99%. Further details of the FMI radar network and processing routines are described in Saltikoff et al. (2010).

The basic radar volume scan consists of thirteen PPI sweeps. The FMI operated LAPS version (hereafter FMI-LAPS) is using the six lowest elevations: 0.3 (alternative 0.1 or 0.5 depending on site location), 0.7, 1.5, 3.0, 5.0 and 9.0, which are scanned out to 250 km, and repeated every 5 minutes. These radar volume scans are further used in LAPS routines for the rain-rate calculations but also, as proxy data to the Lightning Data Assimilation (LDA) method (see Sect. 3.2).

## 2.3 The Lightning Location System (LLS)

The Lightning Location System (LLS) of FMI is part of the Nordic Lightning Information System (NORDLIS). The system detects cloud-to-ground (CG) and intracloud (IC) strokes in the low-frequency (LF) domain. Finland is situated between 60–70°N and 19–32°E and thunderstorm season begins usually in May and lasts until September. During the period 1960–2007, on average, 140'000 ground flashes occurred during approximately 100 days per year (Tuomi and Mäkelä, 2008). The present modern lightning location system (LLS) was installed in summer 1997 (Tuomi and Mäkelä, 2007; Mäkelä et al., 2010; Mäkelä et al., 2016). The system consists of Vaisala Inc. sensors of various generations, and the sensor locations in 2015 and the efficient network coverage area can be seen in Fig. 2. Lightning location sensors detect the electromagnetic (EM) signals emitted by lightning return strokes, measure the signal azimuth and exact time (GPS). Sensors send these information to the central processing computer in real time which combines them, optimises the most probable strike point and

outputs this information to the end user. More detailed information of LLS principles are described in Cummins et al. (1998).

**2.4 Verification periods**

The verification periods are limited to summer season (the active convective season in Finland) where two long
periods were included into the verification; a) 1 April to 1 September, 2015 and b) 1 May to 26 July from 2016. These long verification periods include many cases of stratiform precipitation with no lightning and therefore, the effective impact by lightning is diluted (e.g. no influence by the LDA-method). Hence, two subsets of 25 lightning intense cases (e.g. situations with heavy rain and strong convection), datasets c) and d), were used to explicitly find the lightning induced impacts. The dataset c) includes full days (24h periods) with more than 100
CG strokes per day. The dataset d) includes only the stations and time-intervals affected by lightning (defined as stations with maximum distance of 30 km to the lightning position and within the 1 hour accumulation time-interval, hereafter called scaled dataset). An early dataset from 2014, dataset e), consist of four days (03, 23, 24 and 30 of July, 2014), also with more than 100 CG strokes per day. This dataset was used to perform several autonomous experiments with the FMI-LAPS LDA system, in the early stage of the development of the LDA-
method.

**3 Methods**

The systems used to assimilate radar, gauge and lightning measurements are described in Sect. 3.1-3.2. The impact of different integration time periods on the Regression and Barnes (RandB)-method is shown in Sect. 3.3-3.4 and, finally, the verification methods in Sect. 3.5.

**3.1 The Local Analysis and Prediction System (LAPS)**

The LAPS produces 3D analysis fields of several different weather parameters (Albers et al., 1996). LAPS performs a high-resolution spatial analysis where observational input, from several sources, are fitted to a coarser background model first-guess field (e.g. ECMWF forecast model). Additionally, high resolution topographical data are used when creating the final analysis fields. The FMI-LAPS products are mainly used for
now-casting purposes (i.e. what is currently happening and what will happen in the next few hours), which is of critical interest for end-users who demand near real-time products.

The FMI-LAPS use a pressure coordinate system including 44 vertical levels distributed with a higher resolution (e.g. 10 hPa) at lower altitudes and decreasing with height. The horizontal resolution is 3 kilometres and the temporal resolution is 1 hour. The domain used in this article covers the whole Finland and some parts of the
neighbouring countries (Fig. 1b). LAPS highly relies on the existence of high-resolution observational network, in both space and time, and especially on remote sensing data. The FMI-LAPS is able to process several types of in-situ and remotely sensed observations (Koskinen et al., 2011), among which radar reflectivity, weighting gauges and road weather observations are used for calculating the precipitation accumulation. The Finnish radar volume scans are read into LAPS as NetCDF format files, thereafter the data is remapped to LAPS internal

Cartesian grid and the mosaic process combines data of the different radar stations (Albers et al., 1996). The rain-rates are calculated from the lowest levels of the LAPS 3D radar mosaic data, via the standard Z-R formula (Marshall and Palmer, 1948), which is then used for precipitation accumulation calculations (see Sect. 3.2). Other information on observational usage, first-guess fields, the coordinate system etc. is described in Gregow et al. (2013).

In this study the lightning data are ingested into the FMI-LAPS. Modifications have been made to the software, in order to use it together with FMI operational radar input data and the new lightning algoritms.

**3.2 The LAPS Lightning Data Assimilation (LDA) method**

A Lightning Data Assimilation (hereafter LDA) system has been developed by Vaisala and distributed as open and free software (Pessi and Albers, 2014). The LDA-method is constructed to build up statistical relationships
between radar and lightning measurements. The lightning information used for the LAPS LDA-method is the location data (e.g. time, longitude and latitude) for each CG lightning stroke. LDA counts the amount of CG lightning strokes and converts lightning rates into vertical radar reflectivity profiles, within each LAPS grid-cell. The radar reflectivity-lightning (hereafter Rad-Lig) relationship profiles may differ depending on the local geographical regime and climate. A set of default profiles are included within the LDA package, profiles that
were derived over the eastern United States with the use of radar data from NEXRAD network and lightning data from GLD360 network (Pessi, 2013 and Said et al., 2010). These profiles can be used as a first guess, if profiles for the local climate are not available.

For this study over Finland, climatological Rad-Lig reflectivity relationship profiles were estimated using NORDLIS-LLS lightning information and operational radar volume data from Finland area, during summer
2014. A total of approximately 220'000 lightning strokes were used for this calibration. The FMI-LAPS LDA is using 5 minutes interval of lightning- and radar data, within a LAPS grid-box of resolution 3*3 km. The collected strokes are divided into binned categories using an exponential division (i.e. $2^n...2^{n+1}$), following the same method used in Pessi (2013). This results in 6 different lightning categories (e.g. with 1, 2-3, 4-7, 8-15, 16-31 and 32-63 strokes) for the NORDLIS-LLS dataset. For each of these 6 categories the average reflectivity is
calculated at each grid-point, for each level, and gives the Average Rad-Lig profiles (Fig. 3a), which is the baseline method. There is a good correlation ($R^2$=0.95) between the maximum reflectivity of profile and number of lightning strokes (Fig. 3b; results shown for the Average Rad-Lig profiles). We extend this method to also calculate the 3'rd Quartile (i.e. 75%-percentile) and the Variable Quartile Rad-Lig profiles, for each category. The Variable Quartile method uses a range between 50%-percentile (for the lower dBZ values) up to the 95%-
percentile (for the highest dBZ values). The specific percentiles used for the 6 categories are 50-, 50-, 60-, 75-, 90- and 95% percentiles, respectively. The reason is to take into account the uncertainties in the low categories (due to larger spread and bias in the collected datasets) and on the other hand, rely on the high percentiles for the high categories (since these have less spread). The profiles from the two categories with largest amount of strokes have the least data, because they are the rarest categories. All datasets suffer from missing data at some
height levels, but these two categories are more sensitive, due to the overall small data amounts. This can sometimes create artificial peaks of too low reflectivity values. This was especially seen at high altitudes, which

can partly be explained by the radar measurement geometry. Therefore these two reflectivity profiles have been manually smoothened to have the same shape as the other profiles.

The Rad-Lig reflectivity profiles can be used either independently, or merged with the radar data, in the LAPS accumulation analysis. When merging the two sources, radar and lightning reflectivity values are compared at each grid-point, both horizontally and vertically. The data source giving the highest reflectivity value will be used in that LAPS grid-point. The logic behind this is that the radars are more likely to underestimate, than overestimate the precipitation (due to attenuation, beam blocking or the nearest radar missing from network; e.g. Battan, 1973 and Germann, 1999), especially in thunderstorm situations. This is an approximation, aiming to compensate for the most serious radar error sources, which could be subject for further improvement in future developments (especially if independent quality estimates of the radar data become available). LAPS then uses the generated 3D volume reflectivity field in a similar manner as it would use the regular volume radar data, for example, to adjust hydrometeor fields and rainfall.

The reflectivity ($Z$; $mm^6/m^3$) parameter measured by the radar, or estimated by LDA-method, is converted to precipitation intensity ($R$; mm/h) within LAPS, using a pre-selected Z-R equation (Marshall and Palmer, 1948) as of the type:

$$Z = A \cdot R^b \quad .$$
(3)

Where A and b are empirical factors describing the shape and size distribution of the hydro-meteors. In FMI-LAPS's implementation A=315 and b=1.5 for liquid precipitation, which is relevant in this study carried out during summer period. These static values introduce a gross simplification, since the drop size and particle shapes vary according to weather situation (drizzle/convective, wet snow/snow grain). Challenging situations include both convective showers, with heavy rainfall, and the opposite case of drizzle, with little precipitation (Uijlenhoet, 2008). On the other hand, the same static factors have been used for many years in FMI's other operational radar products, and looking at long-term averages, the radar accumulation data does match the gauge accumulation values within reasonable accuracy (Aaltonen et al., 2008). The intensity field ($R$; Eq. 3) is calculated at every 5 minutes and the 1 hour accumulation is thereafter obtained accumulating 5 minutes intervals. Gires et al (2014) have shown that the scale difference has an effect in verification measures (such as normalized bias, e.g. RMSE) but it decreases with growing accumulation time (e.g. from 5 to 60 minutes). In our study, the 60 minutes accumulation period is smoothing some of the differences.

The following FMI-LAPS precipitation accumulation products are calculated based on Radar- (hereafter Rad_Accum), LDA- (hereafter LDA_Accum) and the combined radar and LDA- (hereafter Rad_LDA_Accum) precipitation accumulation.

### 3.3 The FMI-LAPS Regression and Barnes (RandB) analysis method

The FMI-LAPS RandB-method corrects the precipitation accumulation estimates using radar and gauges datasets. The first step in this method is to make the radar-gauge correction using the Regression method. Data of hourly accumulation values are derived from the gauge-radar pairs within the LAPS grid (i.e. from same location and time), and from this a linear regression function can be established. The corrections from Regression method is applied to the whole radar accumulation field and thereafter used as input for the second step, the Barnes analysis. Within LAPS routines the Barnes interpolation converge the radar field towards gauge accumulation

measurements at smaller areas (i.e. for gauge station surroundings). Several iterative correction steps are performed within the Barnes analysis, adjusting the final accumulation. The FMI-LAPS RandB-method is described in more details in Gregow et al. (2013).

In this article, the RandB-method is used to calculate the precipitation accumulation with the use of radar, gauges, lightning and the combination of radar-lightning. This gives the additional three FMI-LAPS accmulation

products: Rad_RandB, LDA_RandB and Rad_LDA_RandB, respectively.

**3.4 RandB-method and the integration time period**

The original FMI-LAPS RandB-method uses radar and gauge data from the recent hour. Using only the latest hour, the gauge observational dataset can suffer from too few observations and thereby affect to the quality and robustness of the Regression- and Barnes calculations. As a further investigation in this article we use a selection

of longer time periods (e.g. the previous 6, 12, 24 hours and 7 days of data) in order to build up a larger radar-gauge dataset. These datasets are thereafter used to make the correction within the RandB-method.

We have limited our studies to compare how the occurring synoptic weather situation, i.e. frontal or convective situation (1 to 12 hours), and the medium time-range information (24 hours to 7 days) impact on the accumulation analysis. The longer integration time, the less information on the situational weather occurring at

analysis time, i.e. the dataset is getting more smoothed and extremes might disappear.

Verification was done for the summer period 2015, using the input from radar and lightning, and gives the following resulting accumulation products: Rad_LDA_RandB (i.e. dataset collected within the last 1 hour), Rad_LDA_RandB_6hr, Rad_LDA_RandB_12hr, Rad_LDA_RandB_24hr and Rad_LDA_RandB_7d, respectively.

**3.5 Verification methods**

The hourly accumuation results have been verified against surface gauge observations, both dependent and independent stations. The dependent station data are included into the FMI-LAPS analysis calculating the 1 hour precipitation accumulation, i.e. the analysis is depending on the station information used as input. There are 7 independent stations which are excluded from the LAPS analysis. Note that in the Rad_Accum and

Rad_LDA_Accum products the gauge data has not been used, therefore all gauge stations are independent references for their verification. In this study we apply a filter to the verification datasets, where hourly accumulation data less than 0.3 mm are discarded (due to the lowest threshold value of surface gauge measurements from FMI database). In a separate verification exercise for the 2016 data, only stations located more than 100 km and more than 150 km from nearest radar station were used, to demonstrate the potentially

detoriating quality of radar data with distance to the radar due to e.g. attenuation and beam broadening (1 degree beam is 5 km wide at distance of 250 km).

The validation of the different analysis methods are based on the logarithmic standard deviation (STDEV; Eq. 4), root-mean-square deviation (RMSE; Eq. 5), and Pearson's correlation coefficient (CORR; Eq. 6):

$$STDEV = \frac{1}{N-1} \sum_{i=1}^{N} \left( \log\left(\frac{Analysis}{Gauge}\right)_i - \overline{\log\left(\frac{Analysis}{Gauge}\right)} \right)^2$$

(4)

$$RMSE = \sqrt{\dfrac{\overline{\sum_{i=1}^{N}\left|(Analysis - Gauge)_i\right|^2}}{N-1}}$$

(5)

$$CORR = \dfrac{\sum_i \left|\left(Gauge_i - \overline{Gauge}\right)\left(Analysis_i - \overline{Analysis}\right)\right|}{\sqrt{\sum_i \left(Gauge_i - \overline{Gauge}\right)^2 \sum_i \left(Analysis_i - \overline{Analysis}\right)^2}}$$

(6)

STDEV quantifies the amount of variation (i.e. spread) of a dataset. A low STDEV indicates that the data points tend to be close to the mean value of the dataset. Here we use the logarithm of the quotients, in order to get the datasets closer to be normally distributed. RMSE is a quadratic scoring rule, which measures the average magnitude of the error. Since the errors are squared before they are averaged, RMSE gives a relatively high weight to large errors. CORR gives a measure of the linear relationship (both strength and direction) between two quantities.

## 4 Results

Verification results using lightning data is presented in Sect. 4.1 and the impact from different integration time intervals in Sect. 4.2.

### 4.1 FMI-LAPS LDA results

The verification for the entire summer of 2015, i.e. using verification dataset a) including days with no thunderstorms, assures that introducing lightning data has no significant impact in the overall performance of the system. The impact by using LDA-mehtod for estimating the precipitation accumulation is neutral, for this long verification period (shown in Fig. 4, where the data are from dependent stations). Same result is seen in the scores of RMSE, STDEV and CORR values (not included here). Since the data have been much influenced by weather situations not relating to lightning, the focus will be on the subsets, i.e. datasets c) and d), the 25 days periods of intense lightning days of both 2015 and 2016, respectively.

The 25 days period with frequent thunderstorms during summer 2015, verification dataset c), for which we used the Average mehtod to calculate the Rad_Lig profiles, shows an inconsistent result using lightning data (see Table 1; left column). For the independent dataset the Rad_LDA_Accum has a slightly improved result (lower RMSE value), when comparing with Rad_Accum. On the other hand, Rad_LDA_RandB get worse results, as can be seen from the RMSE and CORR. The Dependent data show almost neutral impact (RMSE is slightly better for Rad_LDA_RandB), by the use of LDA-method and Average calculated Rad-Lig profiles.

Figure 5 show the results using verification dataset e), where different Rad-Lig profiles are compared (e.g. Average-, 3'rd Quartile- and Variable Quartile profiles) and validated against Rad_Accum. The precipitation accumulation estimates are improved at high accumulation values (> 5 mm), using either 3'rd- or Variable Quartile profiles. Simultaneously, they both add to the overestimate in low accumulation values (< 5 mm). The 3'rd Quartile profiles gives the largest overestimate, over the whole accumulation scale. The Variable Quartile

gives the overall best result, with improved estimates for high accumulation values and only slight overestimation at low values.

The results, from the scaled dataset d) and the dependancy of distance to radar location, reveal the positive impact by using the lightning data as input for the LAPS-LDA model. Hence, using the Variable Quartile profiles in the accumulation analysis for the 25 days dataset of summer 2016 give a positive impact to the accumulation estimates (see Table 1, right column). Even if the improved scores are relatively small (largest reduction in RMSE being 6.3%) the LDA-mehtod show a consistent correction of the results. The independent verification give decreased RMSE and increased CORR values for Rad_LDA_Accum compared to Rad_Accum. Also, Rad_LDA_RandB get smaller errors than Rad_RandB (see STDEV and RMSE in Table 1; most upper right panel). For the dependent stations all scores are improved using LDA-method, especially the RMSE (as seen in Table 1; right column, second panel). The verification of distance dependancies, i.e. for observations further away than 100- and 150 km from nearest radar stations, show improved accumuation estimates when using the LDA-method (see Table 1, right column, two last panels). The RMSE and CORR scores for Rad_LDA_Accum and Rad_LDA_RandB are better than Rad_Accum and Rad_RandB, respectively. Here, only dependent gauges are availabe for verification.

Comparing accumulation results from the 4-days period, i.e. verification dataset e), for radar alone (Rad_Accum; black markers in Fig. 6) and lightning alone (LDA_Accum; red markers in Fig. 6), it is clear that the use of LDA_Accum is less accurate than Radar_Accum results. Figure 6 also show that the Rad_LDA_Accum estimates (using the baseline method, with Average Rad-Lig profiles) are amplified over the whole range of precipitation values, compared to Rad_Accum (Fig. 6; compare the blue with the black markers). For the high accumulation values (> 5 mm/h) this is a positive effect, while in lower range (< 5 mm/h) there is an overestimation of the results.

**4.2 RandB-method and impact from different integration periods**

The plotted results of different time sampling periods are seen in Fig. 7, where the density of points are drawn as isolines in the scatter plot, with verification against the independent stations from verification dataset a). The Rad_LDA_RandB (i.e. using observations from the latest 1 hour) does give the best result, when compared to Rad_LDA_Accum, Rad_LDA_RandB, Rad_LDA_RandB_6hr, Rad_LDA_RandB_12hr, Rad_LDA_RandB_24hr and the Rad_LDA_RandB_7d output. The statistical scores shown in Table 2 also imply the same result. The Rad_LDA_Accum (e.g. a method not using RandB) is included as a reference, when comparing the results of different integration periods.

**5 Discussions and conclusions**

The aim of this article is to describe new methods on how to improve the hourly precipitation accumulation estimates, especially for heavy rainfall events (> 5 mm) and, as much as possible, also for the low-valued ranges (< 5 mm).

The strength of the LDA-method is that the radar and lightning information can be merged and complement each other. This is especially important in areas of poor, or even none, radar coverage, where the lightning information

will improve the reflectivity field and thereby the hourly precipitation accumulation analysis. It is important to recall that in the LAPS accumulation process, the reflectivity field is the first step, which is then corrected with gauges (e.g. if there is no reflectivity field, gauges will not be used and will be no accumulation field). The results in this article are limited to Finland but considering extending this area to include Scandinavia, the LDA-method will become even more useful. There are also other LAPS-users in other parts of the world, whom we want to encourage to continue this work.

The whole summer periods of 2015 and 2016 show neutral impact in the results, using the LDA-method, scores are not included here but Fig. 4 show the graphs for verification dataset a). It is important to make long-term verification in order to see that the system is robust and does not generate any bad data during any weather situation, i.e. sanity-check of system. Though, in order to narrow down our analysis to areas and times where lightning did occur (i.e. exclude stratiform precipitation), we focused our results on the subset of 25 lighting instense days for both 2015 and 2016, datasets c) and d), respectively. The subset of 2015, using the Average method, gave inconsistent results and no unambiguous conclusions could be drawn (Table 1, left column).

New methods to calculate the Rad-Lig profiles were tested and reveal that the Variable Quartile method improves the estimates for the large accumulation (i.e. > 5 mm), though with some overestimation in low accumulation (Fig. 5). The 3'rd Quartile approach gives the highest impact to the whole accumulation field, which results in large overestimates for the low accumulation values (i.e. between 0-5 mm). The Average method smoothens out the small-scale variances, which is observed in heavy convection. Hence, the collected radar reflectivity profiles are less representative and, therefore, the calculated Rad-Lig profiles will have too low values in these cases. As a result, the Average method will give low impact to the final precipitation accumulation estimates, compared to the use of 3'rd Quartile- and Variable Quartile method (Fig. 5). One should also mention that there is an overall uncertainty due to instrumental errors and the collocation between observations, within the LDA-method. This could potentially result in dislocation and bad quality of the received radar- and lightning measurements, which would affect to the calculated Rad-Lig profiles. For example in case of radar attenuation, where strong rainfall weakens some part of the reflectivity field. Here the collected radar profiles will have too low reflectivity values and give underestimated Rad-Lig profiles, especially when using the Average method.

The newest results from 2016 and the 25 days subset shows that there is a benefit using the LDA- (Variable Quartile) method. Mainly all scores are becoming better, few are unchanged, when lightning information is used to estimate the precipitation accmulation (see Table 1; right column). Verifying the dataset with distance to radar stations (i.e. gauges situated further away than 100- and 150 km) also show the same results, the accumulation product is improved with LDA-method. The impact on scores are mainly in the second decimal, but they are consistent, and cleary show the tendency of improvement by using LDA-method with the Variable Quartile profiles. One reason we don't see larger impact by LDA-method could be that the Finnish radar network does have a very high quality and system utilization rate and therefore less impacted by the LDA-method. In upcoming version of FMI-LAPS the verification will be focusing on including areas with poor (or none) radar coverage where gauges are available.

The accumulation products generated from RandB-method are corrected using gauge information. This process is influencing the final accumulation results much more than the contribution from the LDA-method (seen in

Fig. 4 results from dependent dataset, where a, c and b, d panels, respectively, are almost identical). The same result was seen for the independent dataset (not shown here). Even though, we have proven that in case there would be no radar data (for example if the radar is malfunctioning), precipitation accmulation information would be available from lightning data and add value to the final product. This is shown in figure 6, where accumulation would be generated from the LDA-method (as seen in Fig. 6; red markers) and also visualized through the example in Fig. 8, where the radar- and Rad-Lig lowest reflectivity fields are plotted for one analysis time: 16 UTC, 30 July 2014. This case study also demonstrates, how the LDA method can reconstruct the highest reflectivities, but areas with weak precipitation are missing.

In the RandB-method the Regression is used to correct for large-scale multiplicative biases between radar and gauge data. In this article we introduce lightning into the RandB-method, as an additional data source. However, lightning errors are likely to be different from those of radar and gauges and this could have an effect on the methodology used here. In future developments, after collecting longer time series to quantify the nature of uncertainty of lightning-based precipitation estimates, we intend to improve the analysis in this direction.

In the present analysis area we mainly anticipate the usefulness of lightning data as a backup plan of rare but significant cases. For the rare nature of such events it is not possible to collect a statistically representative dataset in a few years: even though attenuation of radar signals or completely missing data is observed several times a summer, it is not so often when such events happen just over a rain gauge station. However, our overall analysis shows that when we include the lightning data every day at every point, it makes in average a small improvement, and it is there as a safety network waiting for the cases where radars fail.

For the near real-time accumulation product, data used from the recent hour of analysis time does give the best precipitation accumulation result (Table 2 and Fig. 7). We see correlation peaking at 1 hour integration period and decreasing already for the 6 hours period. Therefore, according to the result in this study, the use of long time integration periods for the RandB-method (up till 7 days in this case) does not improve the hourly precipitation accumulation analysis. Berndt et al (2014) compared data resolutions from 10 minutes to 6 hours and reported a large improvement in the correlation (10 minutes to 1 hour the correlation increased 0.37 to 0.57). From 1 hour to 6 hours the corresponding increase was 0.57 to 0.62, respectively. In Norway, Abdella and Alfredsen (2010) have shown that the use of average monthly adjustment factors leads to less than optimal results. One could speculate that there is an intermediate choice of temporal resolution that would improve the results in this article. For example, there could be better results using periods of 2 to 5 hours. This has not been investigated in this article but will be considered in future studies.

**Acknowledgements**

We want to thank NOAA ESRL/GSD and Vaisala for their support of LAPS-LDA developments, Marco Gabella for his encouraging words and Asko Huuskonen for helping in the final and critical stage of evaluating the results.

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

**Table 1. Precipitation accumulation results from summer of 2015 (i.e. dataset c); left column) and 2016 (i.e. dataset d); right column), for periods of the 25 intensive lightning days (e.g. > 100 CG strokes/day) during both years. Precipitation results are shown for radar (Rad_Accum) and radar merged with lightning data (Rad_LDA_Accum), together with and without gauge measurements included with RandB-method (Rad_RandB and Rad_LDA_RandB, respectively). In the lowest panels, only data from more than 100 or 150 km from the nearest radar are used. Verification is performed against both independent and dependent stations ie. those used or left out from the gauge analysis.**

| | Summer 2015 (Average scheme) | | | | | Summer 2016 (Variable Quartile scheme) | | | |
|---|---|---|---|---|---|---|---|---|---|
| **Independent** | Rad_ Accum | Rad_LDA_ Accum | Rad_ RandB | Rad_LDA_ RandB | **Independent** | Rad_ Accum | Rad_LDA_ Accum | Rad_ RandB | Rad_LDA_ RandB |
| Nr Obs | 3206 | 3332 | 256 | 256 | Nr Obs | 1320 | 1333 | 74 | 74 |
| STDEV | 0.27 | 0.27 | 0.11 | 0.11 | STDEV | 0.32 | 0.32 | 0.12 | 0.11 |
| RMSE | 1.66 | 1.64 | 0.58 | 0.70 | RMSE | 2.62 | 2.60 | 0.92 | 0.89 |
| CORR | 0.67 | 0.67 | 0.97 | 0.96 | CORR | 0.64 | 0.65 | 0.96 | 0.96 |
| **Dependent** | | | | | **Dependent** | | | | |
| Nr Obs | | | 3566 | 3567 | Nr Obs | | | 1364 | 1376 |
| STDEV | | | 0.12 | 0.12 | STDEV | | | 0.14 | 0.13 |
| RMSE | | | 0.77 | 0.76 | RMSE | | | 1.27 | 1.19 |
| CORR | | | 0.93 | 0.93 | CORR | | | 0.93 | 0.94 |
| **>100 km** | | | | | **>100 km** | | | | |
| Nr Obs | | | | | Nr Obs | 656 | 656 | 694 | 698 |
| STDEV | | | | | STDEV | 0.34 | 0.34 | 0.15 | 0.15 |
| RMSE | | | | | RMSE | 2.44 | 2.39 | 1.03 | 1.01 |
| CORR | | | | | CORR | 0.66 | 0.67 | 0.95 | 0.95 |
| **>150 km** | | | | | **>150 km** | | | | |
| Nr Obs | | | | | Nr Obs | 153 | 153 | 168 | 171 |
| STDEV | | | | | STDEV | 0.39 | 0.39 | 0.20 | 0.20 |
| RMSE | | | | | RMSE | 2.46 | 2.42 | 1.47 | 1.43 |
| CORR | | | | | CORR | 0.33 | 0.35 | 0.80 | 0.81 |

**Table 2. Impact of the integration time length on RandB-method, for the dependent and independent stations datasets during summer 2015, i.e. dataset a). The Rad_LDA_Accum (e.g. a method not using RandB) is included as an reference.**

| | Rad_LDA_ Accum | Rad_LDA_ RandB _1hr | Rad_LDA_ RandB _6hr | Rad_LDA_ RandB_12hr | Rad_LDA_ RandB_24hr | Rad_LDA_ RandB_ 7d |
|---|---|---|---|---|---|---|
| **Dependent** | | | | | | |
| Nr of observations | 13200 | 16311 | 10956 | 10917 | 10915 | 11033 |
| STDEV (log(R/G)) | 0.25 | 0.13 | 0.13 | 0.13 | 0.14 | 0.14 |
| RMSE | 1.20 | 0.52 | 0.67 | 0.71 | 0.72 | 0.72 |
| CORR | 0.64 | 0.93 | 0.91 | 0.90 | 0.89 | 0.89 |
| | | | | | | |
| **Independent** | | | | | | |
| Nr of observations | 1177 | 1492 | 1028 | 1013 | 1005 | 1014 |
| STDEV (log(R/G)) | 0.25 | 0.15 | 0.22 | 0.22 | 0.22 | 0.22 |
| RMSE | 1.38 | 0.68 | 1.16 | 1.23 | 1.24 | 1.24 |
| CORR | 0.39 | 0.92 | 0.79 | 0.77 | 0.77 | 0.77 |

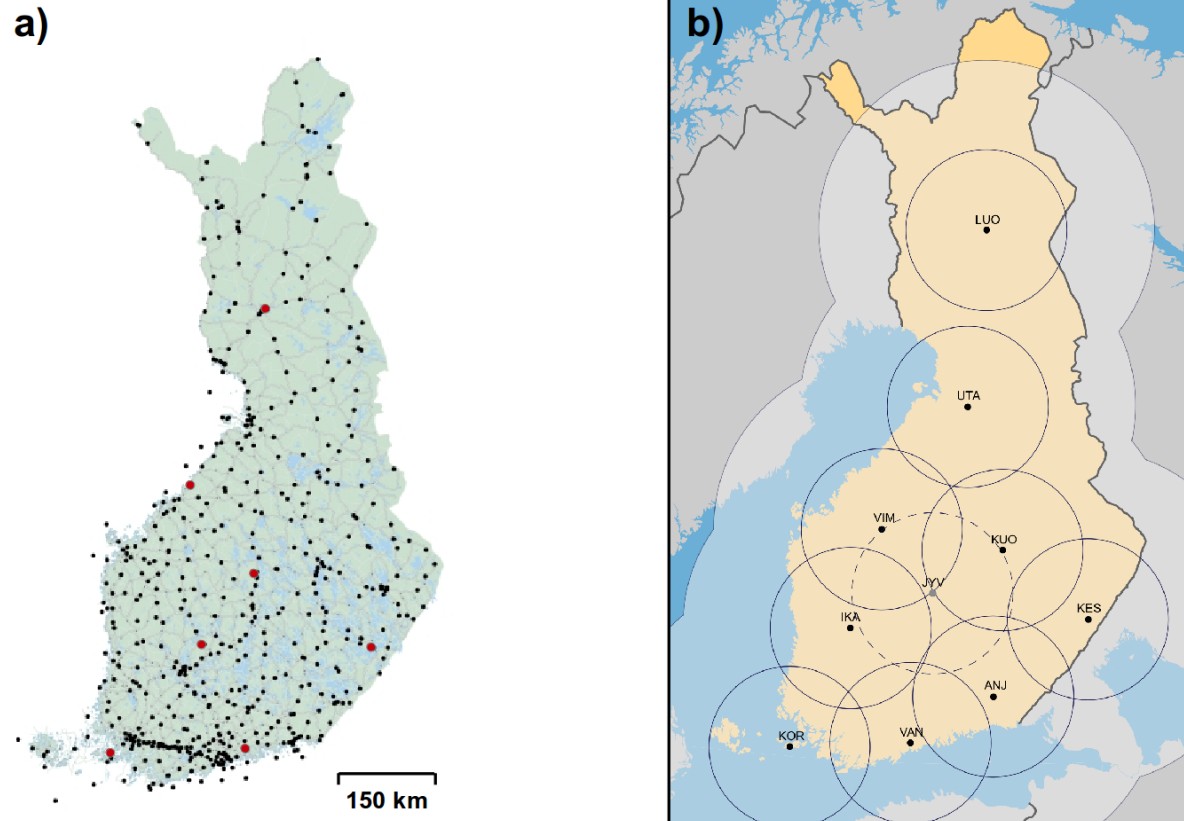

**Figure 1. In (a) the Finnish surface gauge stations are shown (as dots on the map), these are used to measure the hourly precipitation accumulation. The red dots indicate the position of the 7 independent stations used for the verification. In (b) the outer rectangular frame of the map depicts the LAPS analysis domain. The black dots represent the 10 Finnish radar stations and the outer, black curved lines display their coverage. The thin circles surrounding each radar represent the areas where measurements are performed below 2 km height. Dashed circle indicates radar station JYV, which was not included in the radar network during summer 2015.**

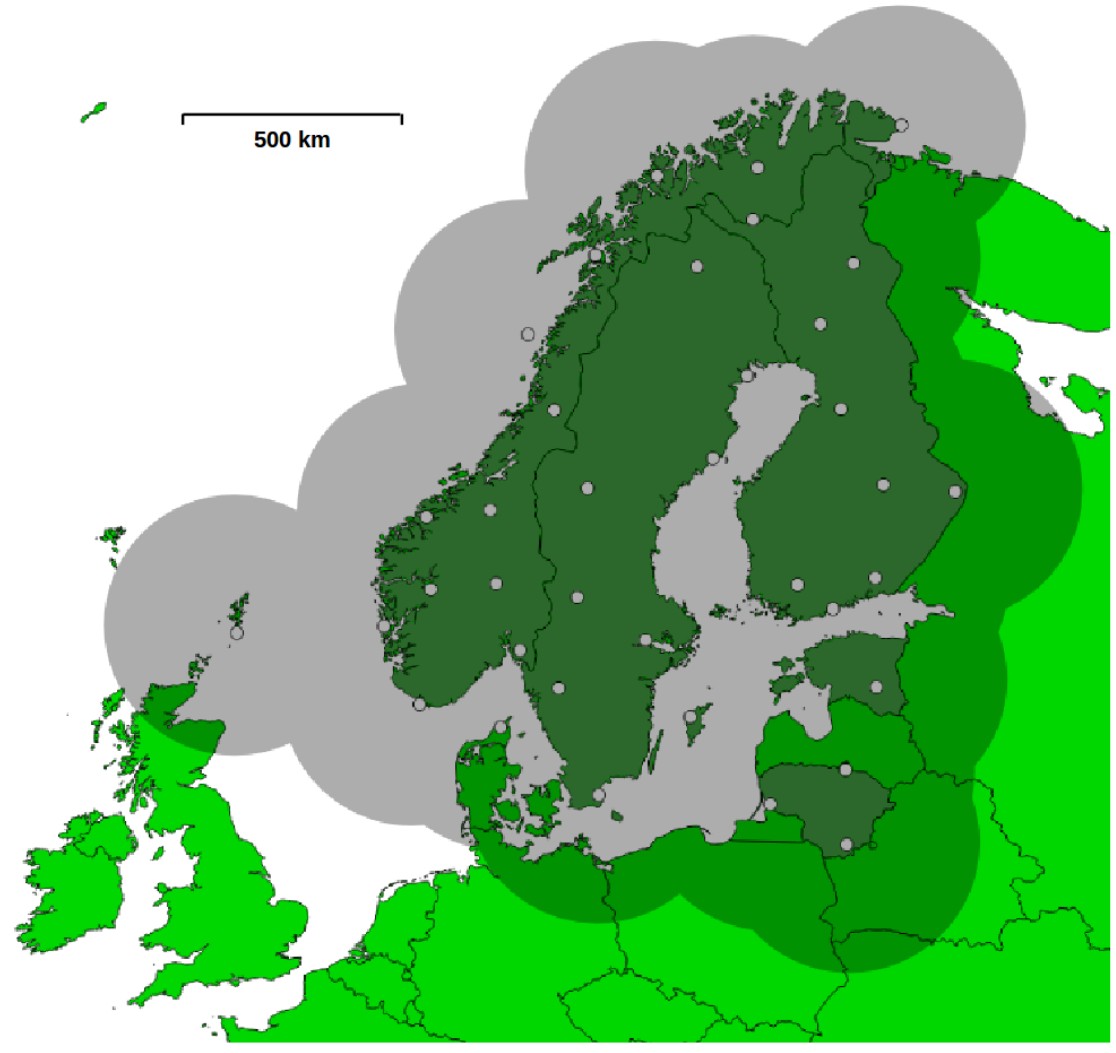

**Figure 2. The LLS sensor locations (white dots) and coverage (grey circular areas), as of year 2015.**

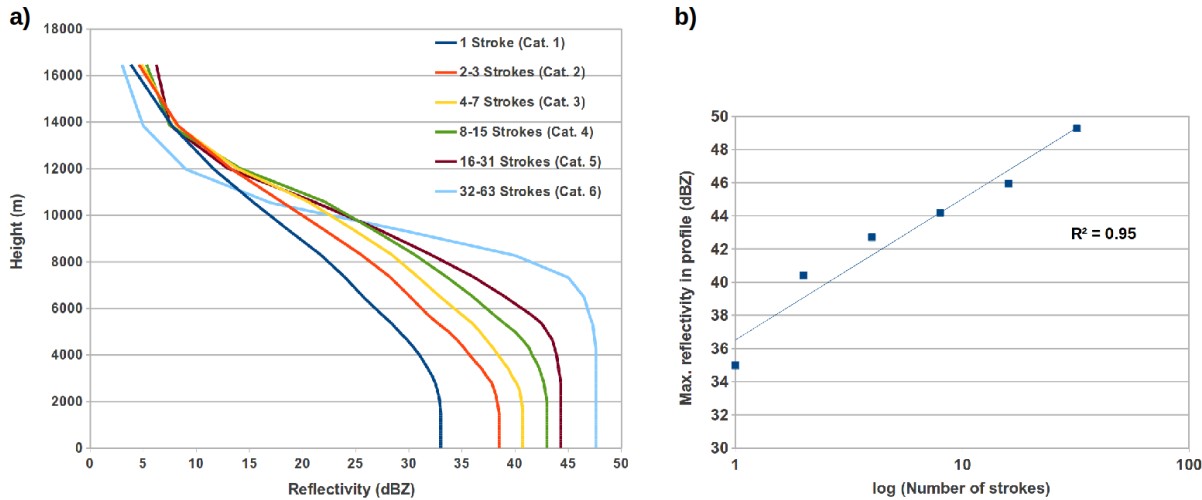

**Figure 3. In a) Rad-Lig relationship profiles (smoothed) from Finland NORDLIS-LLS, calculated using dataset from summer 2014. Profiles are divided into binned categories of strokes, with temporal resolution of 5 minutes and spatial resolution of 3 km. In b) profile's max reflectivity values versus lightning rate (logarithmic-scale of bins).**

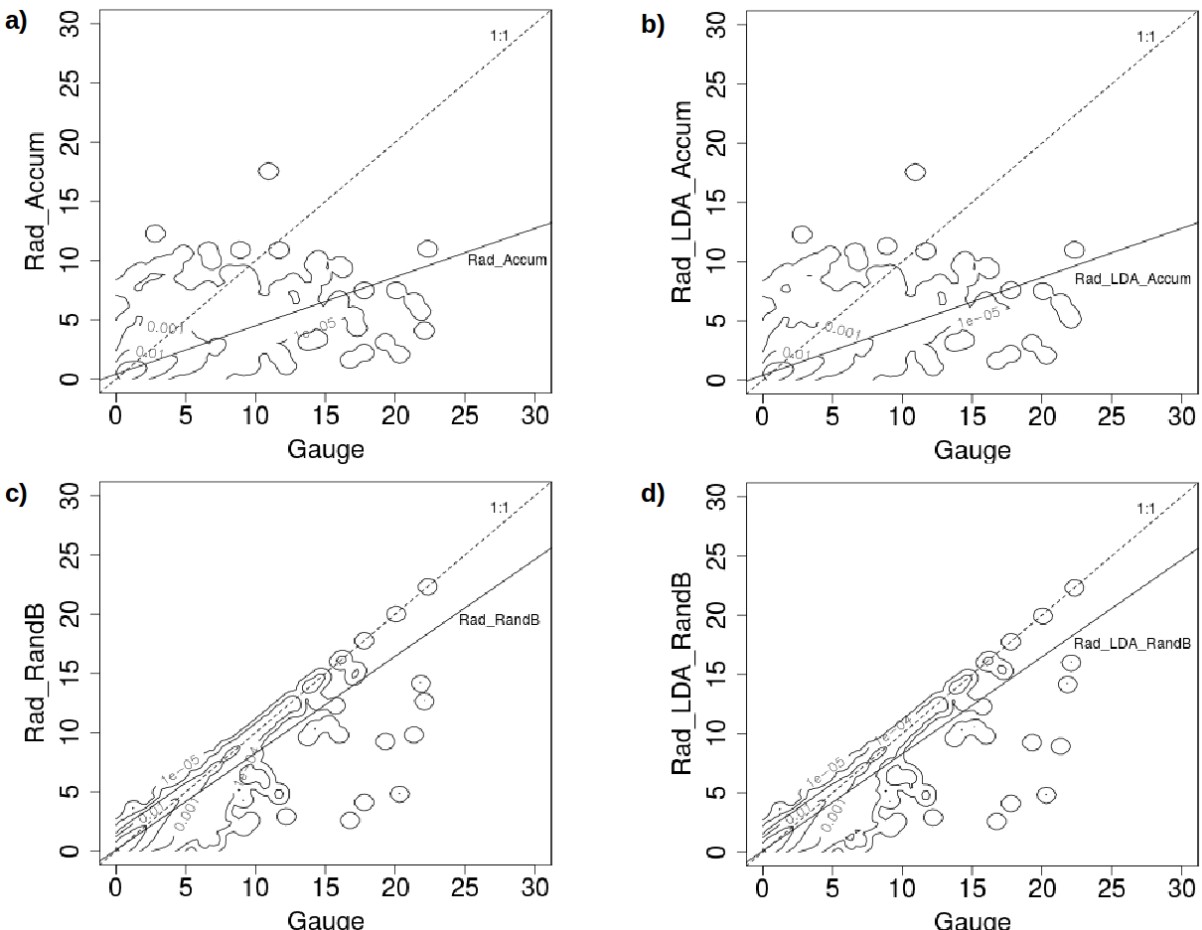

**Figure 4. The FMI-LAPS precipitation accumulation (described in plots with density iso-lines of hourly accumulation values in mm) calculated using 4 different methods. Fit in solid line (see regression equations), perfect solution would align on the 1:1-dashed line. Figure a) Rad_Accum (y=0.410x+0.398), b) Rad_LDA_Accum (y=0.413x+0.396), c) Rad_RandB (y=0.817x+0.093) and d) Rad_LDA_RandB (y=0.819x+0.091). Results are from the dependent gauge dataset during summer 2015, i.e. verification dataset a).**

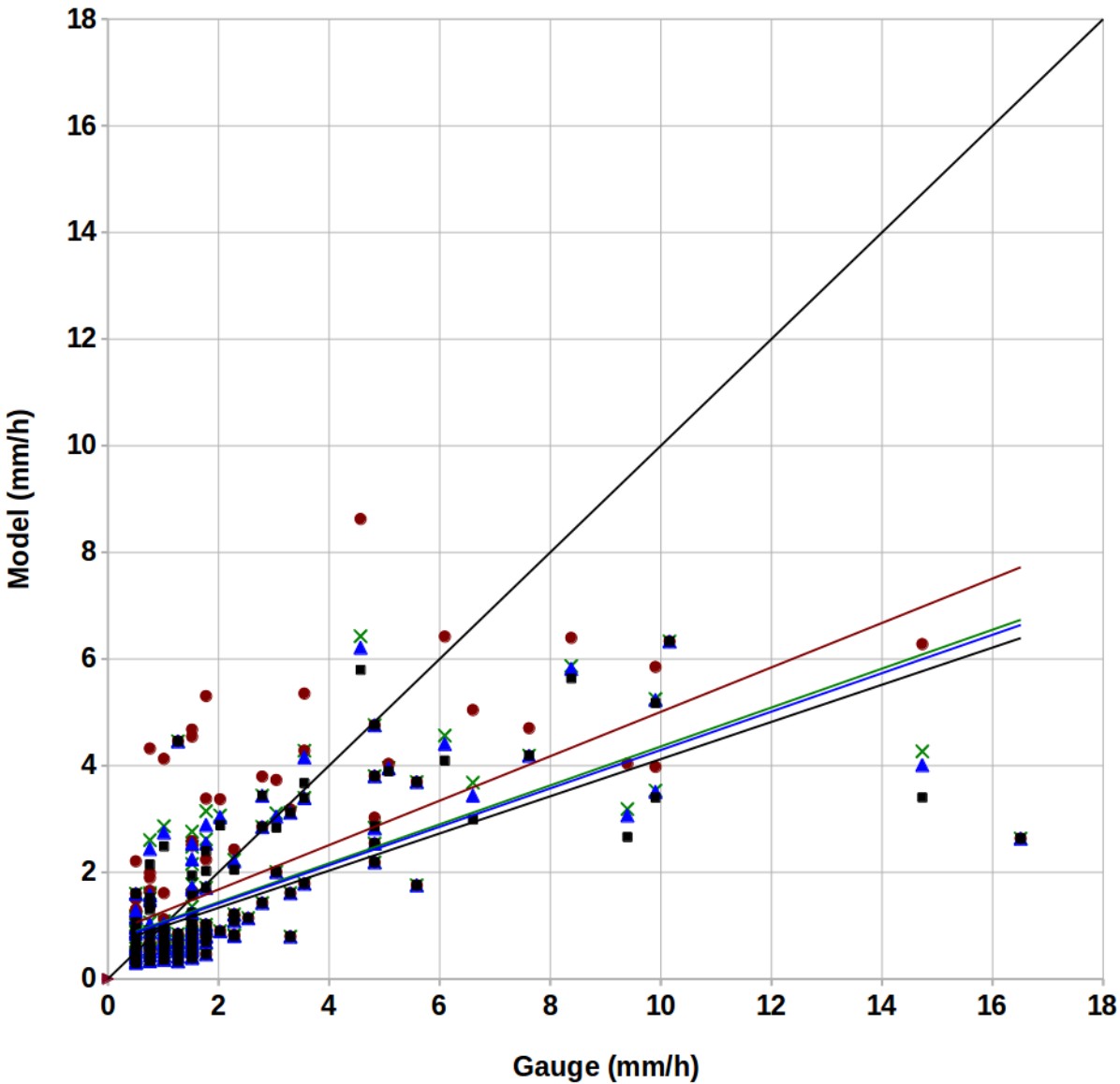

**Figure 5. Verification of hourly accumulation values for Rad_Accum (black squares, with regression line equation y=0.349x+0.638) and LDA_Accum (triangle-, cross- and circular markers), using 3 different methods to calculate the relationship profiles: Average- (blue triangles, y=0.360x+0.691), 3'rd Quartile- (red circles, y=0.417x+0.844) and the Variable Quartile (green crosses, y=0.365x+0.710) accumulation estimates. The corresponding regression lines (see equations) are represented with same color as the markers, for each method. Data are for the 4-days period in summer 2014, i.e. verification dataset e). The best fit curve (i.e. the 1:1 fit) is shown as black solid line.**

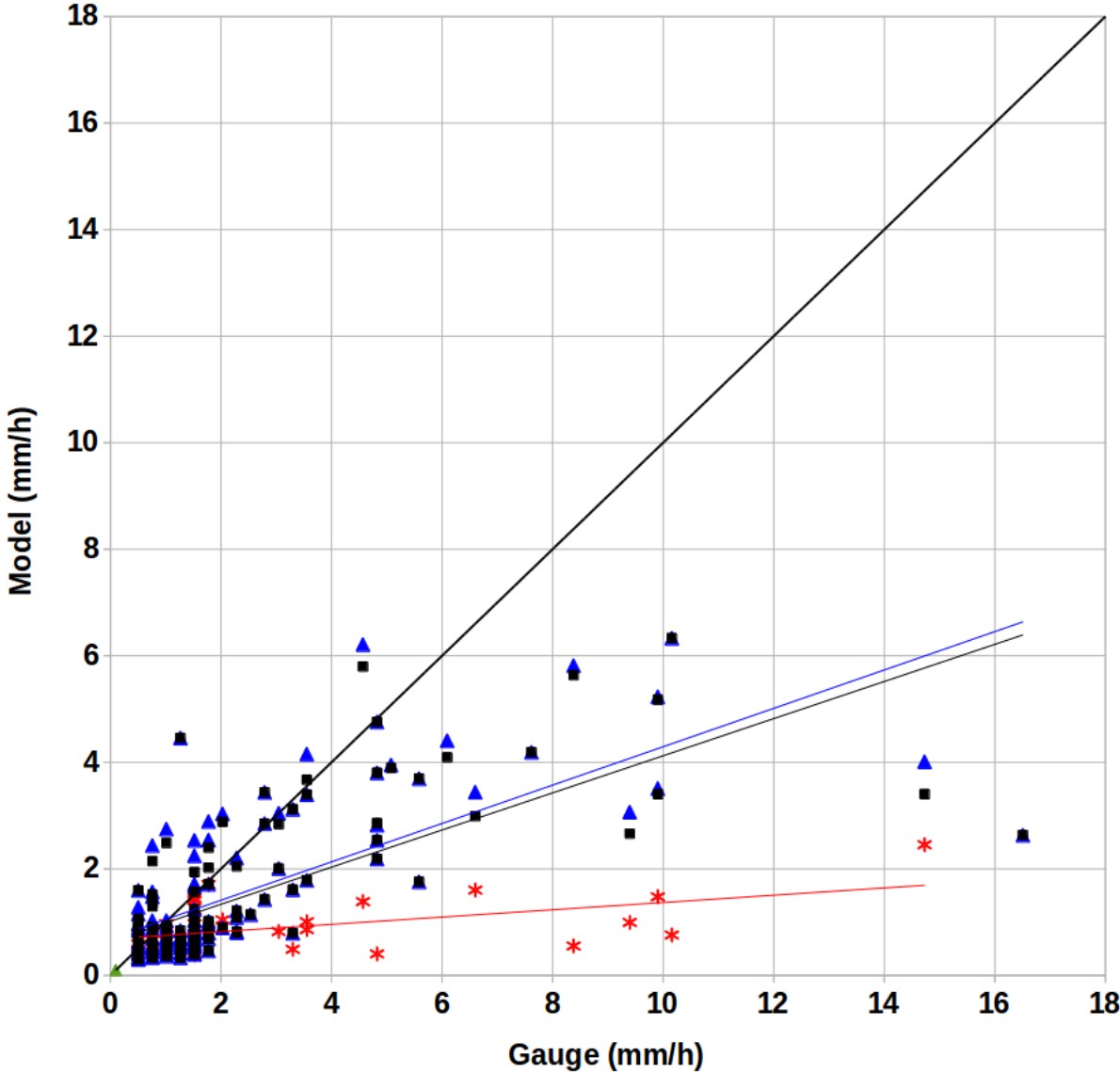

**Figure 6. Verification of hourly accumulation values for LDA_Accum (red stars, with regression line equation y=0.068x+0.685) and the merged Rad_LDA_Accum (blue triangles, y=0.360x+0.691), compared to Rad_Accum (black boxes, y=0.349x+0.638). The corresponding regression lines (see equations) are represented with same color as the markers, for each method. Data are for the 4-days period in summer 2014, i.e. verification dataset e). Black solid line is the best fit line (1:1 fit).**

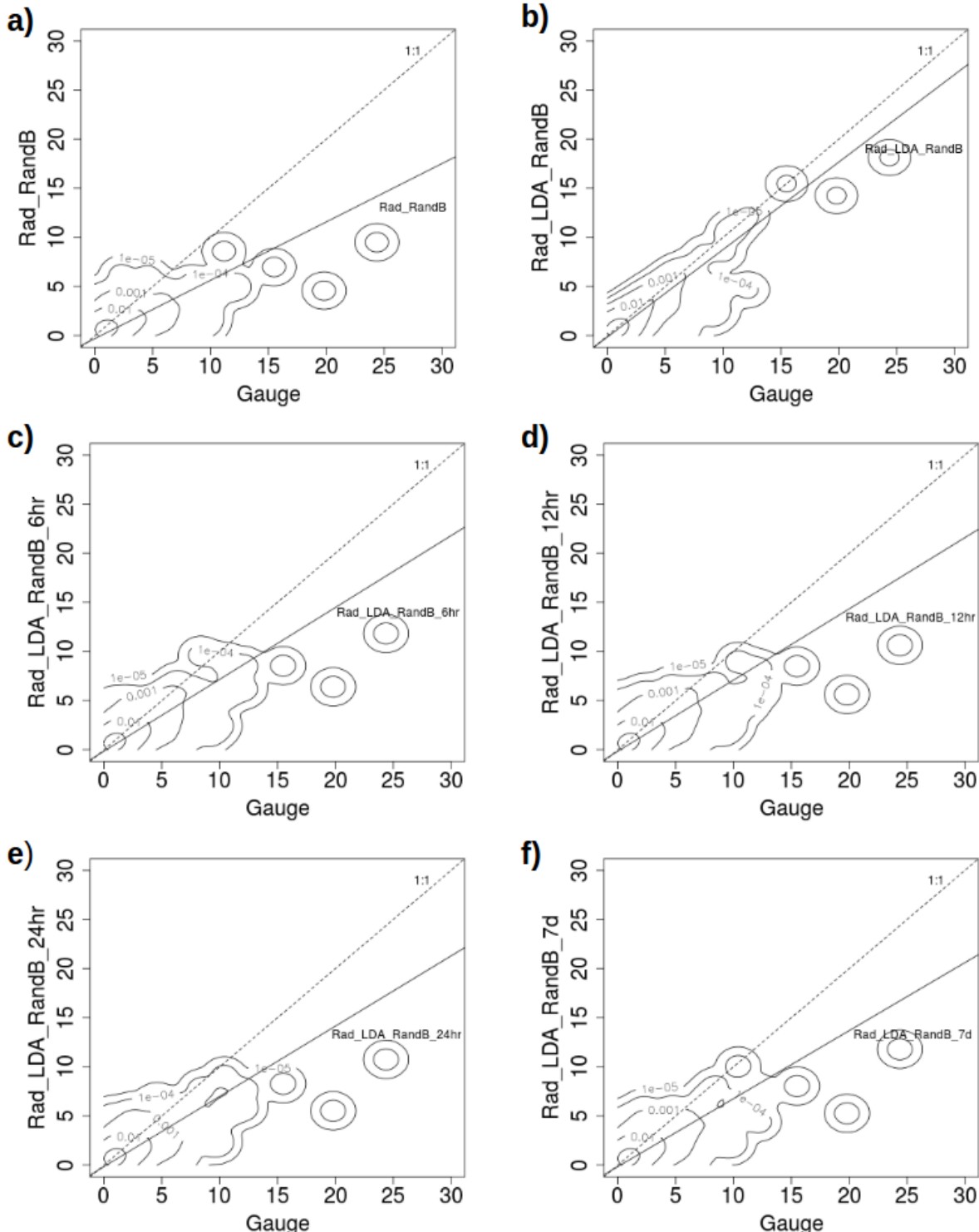

**Figure 7. Impact of changing the integration time length, with verification for the independent gauges, using verification dataset a) from summer 2015. Accumulation plots with density iso-lines of hourly values in mm: a) Rad_LDA_Accum (with regression line equation y=0.594x-0.312), b) Rad_LDA_RandB- (y=0.891x-0.147), c) Rad_LDA_RandB_6hr- (y=0.732x-0.160), d) Rad_LDA_RandB_12hr- (y=0.725x-0.169), e) Rad_LDA_RandB_24hr- (y=0.715x-0.167) and f) Rad_LDA_RandB_7d (y=0.692x-0.166). Fit in solid lines (see regression equations), perfect solution would align on the 1:1-dashed line.**

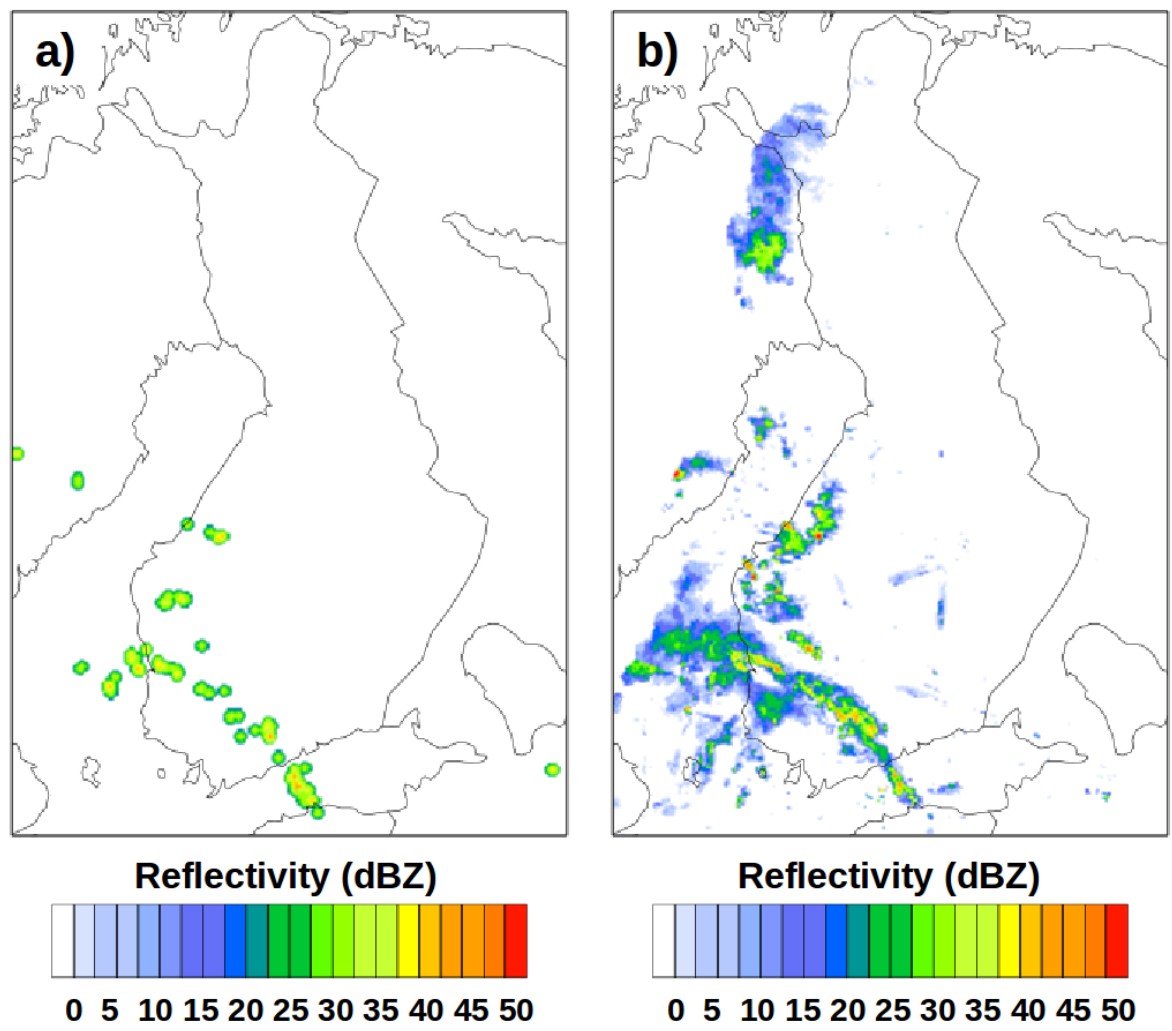

**Figure 8.** Reflectivity field simulated from lightning data alone (left) and, for verification, from radar data alone (right) 30 July 2014 at 16 UTC. Reflectivity color scale is shown below plots.