# Peer review of "Improving the precipitation accumulation analysis using lightning measurements and different integration periods"

_Hydrology and Earth System Sciences, 2016_

## Referee Comment (RC1) · Anonymous Referee #1 · 17 Apr 2016

The aim of the paper is twofold: (i) present and assess a novel operational methodology to include lightning information in radar-gauge precipitation accumulations and (ii) analyze the impact of different integration time intervals in the radar-gauge correction method.

The topic of the paper is of interest for the readers of the journal and the manuscript is well written and concise. The idea of including lightning information in precipitation estimation for intense events is challenging and very interesting both from operational and research points of view.

Nevertheless, the methodology used for the assessment of the new method is not adequate to the purposes of the method and masks out any improvement provided

by the method itself, that, as is currently presented, looks almost useless. For this reason I recommend that the study undergoes a major revision before publication. In the following my major concerns and a list of minor comments.

Major comments:

1. The phenomenon of lightning is usually associated to convection, that is generally characterized by relatively small spatial scales. Such meteorological events are known to be difficult in terms of quantitative precipitation estimation (QPE) because: (i) owing to their small spatial scales are difficult to be adequately sampled by gauges and (ii) radar system may experience important problems due to attenuation of the signal, hail contamination and other issues. Therefore, the use of the LDA potentially represents an important source of information for improving the QPE for such situations. Despite this, results presented in this work show no significant improvement when LDA is used together with the already implemented system (Radar + RandB). If I understood correctly, the information provided by LDA is equivalent to a radar profile of reflectivity corresponding to locations and times in which a lightning occurred. This information is local in terms of space and time (as shown in Fig 6), therefore the potential effects of the use of LDA cannot be detected when large scales (the whole Finland) and long periods (seasonal) are used for the assessment, as they would be masked out. The authors partially recognize this problem and focus on a shorter period (the short, 4-days period) but keep on analyzing the country-scale picture. Furthermore, the use of only 7 independent gauges strongly limits the potential of the study, because of the small scales in which lightning information is available. In fact Tab. 2 confirms this: absolutely no information is available for the short study period (the more interesting one). I would recommend to revise the analysis as follows: (1) limit the analyses, both in space and time, to rainfall events characterized by lightning strikes; (2) select independent gauges in meaningful location for each event.

2. How are Fig 4 and Fig 8 obtained? Are they based on the dependent gauges? Do they show 1h estimates (I assume so since the figures show "mm/h" for the accumulations)? Using 1h estimates for the comparison with the dependent gauges (that are used on 1h scale for the RandB process) will have the Rad_LDA_RandB (1h product) necessarily being the best.

3. I suggest to choose one between r2 and Pareson's correlation coefficient since the two statistics provide the same information. Moreover, basing results on RMSE can be tricky because errors are not weighted.

Minor comments:

1. The title should include more clearly the second objective of the study (impact of different integration time intervals in the radar-gauge correction method)

2. lines 1-5: the sentence is difficult to read. Moreover the second objective of the study should be better stressed. What about: "Two main objectives are addressed: (i) the assimilation of lightning observations in radar and gauge measurements and (ii) the analysis of the impact of different integration time intervals in the radar-gauge correction method."

3. line 6: is the reference Gregow et al. (2013)?

4. The state of art section (lines 28-39) is rather short and can be organized in a clearer way

5. line 47: " usually with a higher quality than radar" a reference can be helpful

6. lines 54-55: "long" rather than "longer", "short" rather than "shorter"

7. line 62: more information about how "poor data quality" stations are identified is needed

8. line 70: Lat-Lon information are not shown in the figure

9. lines 70-72: something is missing in the sentence

10. line 108: I couldn't find the work by Pessi and Albers, 2014

11. lines 120-124: this is not useful for the purposes of the paper

12. line 121 and 127: I couldn't find the work by Pessi, 2013

13. line 184: why 0.3? more details are needed

14. Fig 7: the colors of the regression lines are not explained in the caption

---

## Referee Comment (RC2) · H. Leijnse (Referee) · 17 Apr 2016

This paper describes an assessment of the quality of quantitative rainfall estimates using a combination radar, cloud-to-ground lightning, and rain gauge data. The effect of adding GC lightning data to radar data on rainfall accumulations is investigated, both before and after gauge adjustment. In these analyses, several methods of estimating relations between lightning activity and rain intensity are utilized. Furthermore, the effect of the length of the accumulation interval used for gauge adjustment is also studied. The paper is interesting, and its topic relevant. It is not entirely clear to me what the main goal of this paper is. I think that the paper could benefit from a clearer description of what its main goals are, how the analyses that are presented contribute

to these goals, and coming back to these goals in the Discussions and Conclusions section. When reading the paper for the first time I was sometimes confused because new analyses are proposed in the Results section and some of the methods described in the Methods section were not entirely clear to me. Hence, the paper could benefit from some restructuring, where all methods that are used are presented clearly in the Methods section. I think that the paper needs major revisions in order for it to be suitable for publication. More specific remarks are given below.

**Specific comments**

- 1. Section 1, Given the fact that there are not very many lightning strikes in Finland, how much would you expect that adding this information would influence the final rainfall estimates? I think that this should be thoroughly discussed in the introduction of the paper.
- 2. Section 3.3, It's not entirely clear how radar and lightning data are merged to come up with a final rainfall estimate. Did I understand correctly that the number of recorded GC lightning strikes within a LAPS pixel (3x3 km) and in a 5-minute interval are counted. These counts are then related to a vertical reflectivity profile, and subsequently the maximum of the radar reflectivity and the 'lightning reflectivity' are taken at a given height. The rainfall estimate is then based on the lowest data point. In practice, this means that the rainfall estimate is based on the maximum of the lowest-level radar reflectivity and the lowest-level 'lightning reflectivity'. If this is indeed the case, the description of the method to estimate rain rates could be simplified and clarified. If not, I recommend clarifying this section.
- 3. I.137-149, Given the fact that lightning only occurs in convective situations, it would make sense to me if a Z R relation specifically derived for convective

**HESSD**
rain is used wherever lightning is observed. This would be a simple addition to the LDA that could improve results even further.

- 4. I.174-177, the rationale behind the regression part of the RandB method is that radar rainfall estimates often suffer from large-scale multiplicative biases, and that using regression on radar and gauge data can correct for this error. When adding lightning data to radar data, the errors are likely to be very different, and this could have a large effect on the final rainfall estimates. Something similar can be said for the Barnes-part of the RandB method, where the influence of a gauge correction is in general relatively large compared to the area affected by lightning. I therefore strongly suggest to add a discussion of this in the paper.
- 5. I.176-177, What does it mean that Rad\_LDA\_Accum is the reference?
- 6. Section 4, Why are the graphs where rainfall intensities are compared plotted on log-log scales? If the aim is to study the performance of quantitative precipitation estimation algorithms for high intensities (as is stated in the paper), it would make most sense to me if these graphs were plotted using linear axes.
- 7. I.187-188, what exactly is meant by "the averaged (i.e. 50%-percentile) Rad-Lig reflectivity profiles from the LDA-method."? How were these profiles determined, and based on what data? I think this should be discussed in the Methods section.
- 8. I.192-200, I suggest to remove the  $R^2$  statistic, because it is simply the correlation coefficient squared (see Eqs (6) and (7)) and it hence doesn't add any information relative to CORR.
- I.202-207, The panels of Fig. 4 with LDA added (i.e. panels b and d) do not really add any information, as they are extremely similar to panels a and c, respectively. I therefore suggest making a remark in the text about this, and removing either panels a and c, or panels b and d.

**HESSD**
- 10. I.208-212, I would strongly suggest using different gauges for the independent measurements to test whether using LDA improves rain estimates, because this is what I understand the main objective of this paper to be.
- 11. I.209-210, The use of a 25-day subset is introduced here. I suggest introducing this earlier in the paper (the Methods section). And if this subset is used, what is the added value of using the 4-day subset? I think the clarity of the paper would improve if either the 4-day or the 25-day subset is used.
- 12. I.225-238, It's unclear to me how the new profiles are exactly generated. I strongly suggest to include a good description of this in the Methods section (preferably in Section 3.2).
- 13. I.240-245, Why not test sub-hourly scales?

**Minor remarks**

- 1. I.16, replace "such as;" by "such as" (remove semicolon)
- 2. I.17, replace "eceonomically" by "economically"
- 3. I.39, replace "leass" by "less"
- 4. I.41, what is meant by "a timely accurate manner"?
- 5. I.133, replace "resulting from;" by "resulting from" (remove semicolon)
- 6. I.133-134, consider including clutter as an important source of error
- 7. I.144-145, do you mean to say here that convective rain is important for flooding events? I so, I suggest changing "such situations" to "convective events". The
first time I read this sentence I interpreted "such situations" to be the drizzle that is mentioned in the previous sentence.

- 8. I.187, the 50th percentile is not the average, but the median, and it is either the 50th percentile or the 50% quantile. So I suggest replacing "averaged (i.e. 50%-percentile)" by "median (i.e. 50% quantile)."
- 9. I.192, I suggest calling STDEV "relative standard deviation" or "logarithmic standard deviation" to make clear that it is different from a regular standard deviation.

---

## Referee Comment (RC3) · Anonymous Referee #3 · 17 Apr 2016

This paper addresses the question of how to the use of lightning data can help improve rainfall accumulation estimation. The topic is relevant and paper on this issue are welcomed. However the manuscript exhibit severe flaws, and cannot be published in its current form. The modification needed require in-depth modification.

General comments: - Overall the paper is quite difficult to read, many processing techniques are tested on various data sets, presented not at the same time. May be a scheme summarizing the techniques would help. I think the paper should also be organized better. A solution could be to present "data", then methods with a subsection on the various products from the radar, gauge and lightning, and a subsection on how the comparison is performed. - There seems to be a contradiction between the ab-
stract and the content. abstract I.8-9 : "the performed... usefullness of..." and I.201 "The overall result... neutral to positive impact..." and same comment on the dependent sub-set (I.211-212) - The conclusion seems to be that basically when radar data is available lightning data is rather useless. This is already a result that should be stressed (it is already mentioned). That said I have the feeling that the paper could be more interesting by shifting its scope to how to estimate rainfall (locally) from lightning data when no radar data is available (this would correspond to developing more in depth what is done with figure 7). After this analysis, you could practically test the interest by artificially removing some radar data.

Detailed comments:

- The title "radar -, gauge-" formula is not very clear.

1) Introduction - It should be extended to include a state of art section on the actual topic of the paper, i.e. lightning measurement assimilation for rainfall or more generally in meteorology. (ex among many others: Papadopoulos et al. 2005; Morales et al 2003)

2) Observations and instrumentation - p2 I49 : "LAPS" is used but was not defined before in the manuscript - p3 I76: "as as proxy", one "as" should be removed - section 2.2 : how mosaicking is done ? Some more detail on the radar processing should be provided. how dual polarisation is used ? The differences in terms of sampling area between rain gauges and radar are almost not mentioned (I.81). This discussion should at least be expanded because it can have an influence on the standard scores used after. See for example reference such as Jaffrain and Berne 2012; Gires et al. 2014; on this issue

3) Methods - I94 : randB method mentioned but not defined after - section 3.1 : more details on LAPS are needed. Sentences such as "LAPS uses statistical methods to perform high-resolution analysis" are too general for a scientific paper. It is not clear what is the purpose of LAPS and what data is used out of it, and how it is related to the
radar mosaic product of FMI. - Section 3.2: Some indication on the number of lightning strokes used to calibrate the relation should be given. Figure 3.b : the vertical scale should be changed to improve the visibility of the relation. Could you confirm that the temporal resolution is indeed of 5 min - Eq 3 : so the dual pol capacities are not used ? - I.151-152 : when merging radar and lighting data, why choosing the maximum ? When radar data is available, is not radar more reliable than one choice among the 6 different profiles for lightning ? Please justify this choice. - Section 3.4 : why only one sub section ? You have to say at least few words on Rand B method and Barnes analysis. It is very difficult to understand this section with so little explanations.

4) Results and verification - p. 7 is methodology and should be moved in the corresponding section. Please confirm that in eq. 4-7, values are taken at the hourly resolution ? It might be interesting to test other time steps. - I.187-188 : "the avg Radlig reflectivity profile. Please clarify ? - Figure 4 : please clarify what is plotted and how it was obtained. It is also almost not commented in the text. - Figure 5 : again mention time steps used (1h ?). I.220-224 : the figure should be more commented. - I.228-230 : the quantiles mentioned are not clear. - Figure 7 and comments : the change in quantile seems to improve rainfall estimates. Why was not it used in the first place ? - Figure 8 : please comment more the figure...

References: - Extending the Capabilities of High-Frequency Rainfall Estimation from Geostationary-Based Satellite Infrared via a Network of Long-Range Lightning Observations Carlos A. Morales and Emmanouil N. Anagnostou Journal of Hydrometeorology 2003 4:2, 141-159 - Improving Convective Precipitation Forecasting through Assimilation of Regional Lightning Measurements in a Mesoscale Model Anastasios Papadopoulos, Themis G. Chronis, and Emmanouil N. Anagnostou Monthly Weather Review 2005 133:7, 1961-1977 - Gires, A. et al., 2014. Influence of small scale rainfall variability on standard comparison tools between radar and rain gauge data. Atmospheric Research, 138(0): 125-138. - Jaffrain, J. and Berne, A., 2012. Influence of the Subgrid Variability of the Raindrop Size Distribution on Radar Rainfall Estimators. Interactive comment

---

## Referee Comment (RC4) · S. Ochoa Rodriguez (Referee) · 18 Apr 2016

This paper explores the use of cloud-to-ground lightening data for improving radar-based (and raingauge-adjusted radar-based) precipitation estimation, with a focus on high intensity, convective storms. The study includes an evaluation of resulting rainfall estimates at different temporal accumulations.

The topic of the paper is interesting and the results are potentially useful. However, the paper has a number of major flaws which should be addressed before it can be published.

General comments / major flaws:

- The structure of the article is rather unorganised and the description of data, methods and results are often unclear. The description of the way in which lightening and radar data are merged is very unclear (and bits and pieces of the methodology are spread throughout several sections of the paper), descriptions of the methods are included in the data and results sections, amongst other things (more details are provided below).

- The fact that only 7 independent rain gauges are used for evaluation renders the results statistically weak. This issue is so critical that in one of the cases (Table 2), there are simply no data available for evaluation from any of the independent gauges. Why out of 472 available rain gauges would you only select 7 for independent evaluation? I am aware that you also present the results of performance statistics at non-independent rain gauge locations; however, I truly believe that more interesting results could be achieved if either more independent rain gauges were used or if a cross-validation approach were implemented.

- The results, as currently presented, are too vague and far from the objectives initially set in the abstract and introduction. As stated in the title of the paper, you aim at "improving precipitation accumulation analysis using radar, gauge and lightening measurements". Throughout the paper, the focus is mostly on the added value of lightening data, which does not really lead to significant improvements in radar-based QPEs, let alone gauge-based adjusted radar QPEs. One option would be to change the focus (and consequently the title) of the paper to "the added value of lightening measurements" for generation of QPEs. The added value could be more clearly assessed under different scenarios, including with and without radar data available, with / without rain gauge data. From the results you present, the real advantage of lightening data appears to be in cases when radar data are not available (which can occur either because there simply are no weather radars, due to malfunctioning of the radar or to issues such as beam blockage and/or attenuation); this is very valuable and should be made clearer (and, from my point of view, justifies changing the focus of the paper/changing the way in which objectives and results are described).

Detailed comments:

- Introduction: work by the co-authors of the paper is often cited and a proper review of relevant literature on the actual topic of the paper is lacking. The introduction should be extended to include:

o A review on the use of lightening data for QPE generation

o A more in depth and critical discussion of adjustment of radar QPEs. The only comment so far is that "the use of monthly adjustment factors leads to less than optimal results". Several studies have been carried out which have shown a clear advantage of adjusting radar QPEs based on rain gauge data and other sources of information, with corrections implemented at significantly shorter time scales (as compared to the monthly one that you mention). Also, since the focus of your study is mostly on convective storms, it may be worth discussing the performance of merging techniques for convective storms, which is still a problematic issue (see Jewell & Gaussiat, 2015; Wang et al., 2015) and which could be a case in which lightening information could be useful.

o A review and discussion on the topic of the impact of temporal aggregation on precipitation products. The impact of the temporal scale at which adjustments are performed is a key part of this study. This issue has been discussed in a number of papers which should be reviewed and in the light of which the results of the present study should be analysed (e.g. you should discuss how the optimal temporal resolution that you found (1h) compares to that found in previous studies for the case of convective storms). See for example Berndt et al. 2014.

o I would suggest to remove irrelevant information, such as L26-27 (projected annual precipitation in Northern EU) and keep the introduction focused on the topic of the paper.

- Data Section:

o L47-50 include description of methods and should be removed from the data section. I would suggest that you simply say that three data sources are employed in this study and then go on to explain them (without starting to describe the LAPS, which should be done in the methods section).

o Section 2.1 – I would change title to "Ground rain gauges" or would at least include the word "rain gauge" in it.

o Section 2.1, L57-58 is a repetition of L52; try merging these sentences.

o Section 2.3: A bit more details about lightening sensors would be desirable. It would help the reader understand how is it that lightening measurements can be translated into vertical 'radar' rainfall profiles and so on. I am aware that this information can be found in other papers, but I think it would be helpful and interesting for the reader to find a brief description of the sensors here. Just in the same way that you provide a brief overview of radar QPE generation.

o While all but one of the Finnish radars are dual-pol, it appears that dual-pol parameters are not being used for QPE generation. The authors mention that a single Z-R relationship is used, which implies simplification and assumptions about variable dropsize distribution and the like. Such simplification could clearly be avoided were dual-pol parameters used. It should be made clear (in Section 2.2) that dual-pol parameters are not being used at all and the implications of this should be discussed (I reckon that the use of dual-pol parameters would lead to much larger improvements in the quality of radar QPEs than the use lightening information).

- Methods Section: this section is rather unorganised and a thorough re-structuring would be desirable. Some specific comments/suggestions are the following:

o Section 3.1, L97: "... where a dense observational input, from several sources, are fitted to a coarser background model first-guess field". Please indicate which data sources are used.

o Section 3.1, L102: you indicate the spatial resolution of the FMI-LAPS output, but not the temporal resolution. From subsequent sections I gather that the temporal resolution is 5 min, but it should be clearly indicated in Section 3.1.

o Sections 3.2 and 3.3: these two sections present overlapping information and a clear and integrated description of the integration of lightening data with radar data is missing. I suggest merging these two sections and producing a new and clearer description of the merging method.

- Results section:

o Why using coefficient of determination AND Pearson's correlation coefficient? Their only difference is that the correlation coefficient has a sign, so it may be useful to include both in cases where you expect the correlation coefficient to take negative values. Since this is not the case here, the two performance measures provide very similar information. I would suggest to use only one of the two.

o A description of the log STDEV is missing and a justification for using a log-STDEV instead of the non-log STDEV should be included.

o As mentioned above, this section is unorganised and includes a great deal of description of methods, which should be transferred to Section 3. Also, results should be presented in a more concise and assertive manner (comments such as "neutral to positive impact" should be removed). As suggested above, a shift in the focus of the paper could make it easier to describe the results in a more assertive / critical manner.

o A scale bar should be included in Figure 1.

o Why using log scales in figures 4, 5, 7 and 8? I think a normal (linear) scale would be better. A linear scale is normally used in papers on this topic, so readers are used to it. I do not see any added value in using a log scale and I do think that it hinders interpretation of results.

o I suggest using mm to indicate rainfall accumulations (accompanied by the temporal

aggregation scale), instead of using mm/h (which is the unit normally used for intensities).

- Other comments

o Why not work with sub-hourly temporal accumulations? The lifetime of convective cells is often < 1 h. Since the focus of the study is on convective storms and lightening and radar data are available at high temporal resolution (5 min), it would make sense to evaluate sub-hourly scales.

o Misuse of semicolons throughout the paper. The semicolon should be either removed or replaced by a colon. E.g.:

L16: "such as;" (remove semicolon)

L133: "resulting from;" (either remove or change to colon)

L218: "analysis time;" (change to colon) Many others! Please check.

REFERENCES:

Berndt, C., Rabiei, E. & Haberlandt, U. (2014). Geostatistical merging of rain gauge and radar data for high temporal resolutions and various station density scenarios. Journal of Hydrology, 508, 88-101.

Jewell, S. A. & Gaussiat, N. (2015). An assessment of kriging-based rain-gauge-radar merging techniques. Quarterly Journal of the Royal Meteorological Society.

Wang, L.-P., Ochoa-Rodríguez, S., Onof, C. & Willems, P. (2015). Singularity-sensitive gauge-based radar rainfall adjustment methods for urban hydrological applications. Hydrology and Earth System Sciences, 19 (9), 4001-4021.

---

## Author Comment (AC1) · 26 May 2016

The aim of the paper is twofold: (i) present and assess a novel operational methodology to include lightning information in radar-gauge precipitation accumulations and (ii) analyze the impact of different integration time intervals in the radar-gauge correction method.

The topic of the paper is of interest for the readers of the journal and the manuscript is well written and concise. The idea of including lightning information in precipitation estimation for intense events is challenging and very interesting both from operational

and research points of view.

Nevertheless, the methodology used for the assessment of the new method is not adequate to the purposes of the method and masks out any improvement provided by the method itself, that, as is currently presented, looks almost useless. For this reason I recommend that the study undergoes a major revision before publication. In the following my major concerns and a list of minor comments.

AUTHORS: The authors want to thank the reviewer for the professional and thorough revision of this paper. The new updated article version is attached as Supplement (see PDF-file)

Major comments:

1. The phenomenon of lightning is usually associated to convection, that is generally characterized by relatively small spatial scales. Such meteorological events are known to be difficult in terms of quantitative precipitation estimation (QPE) because: (i) owing to their small spatial scales are difficult to be adequately sampled by gauges and (ii) radar system may experience important problems due to attenuation of the signal, hail contamination and other issues. Therefore, the use of the LDA potentially represents an important source of information for improving the QPE for such situations. Despite this, results presented in this work show no significant improvement when LDA is used together with the already implemented system (Radar + RandB). If I understood correctly, the information provided by LDA is equivalent to a radar profile of reflectivity corresponding to locations and times in which a lightning occurred. This information is local in terms of space and time (as shown in Fig 6), therefore the potential effects of the use of LDA cannot be detected when large scales (the whole Finland) and long periods (seasonal) are used for the assessment, as they would be masked out. The authors partially recognize this problem and focus on a shorter period (the short, 4-days period) but keep on analyzing the country-scale picture. Furthermore, the use of only 7 independent gauges strongly limits the potential of the study, because of the small

scales in which lightning information is available. In fact Tab. 2 confirms this: absolutely no information is available for the short study period (the more interesting one). I would recommend to revise the analysis as follows: (1) limit the analyses, both in space and time, to rainfall events characterized by lightning strikes; (2) select independent gauges in meaningful location for each event.

AUTHORS ANSWER: We fully agree to the concerns expressed by the reviewer in above comments. Though, at that time, setting up this system during 2014-2015, this was the best we could do due to many reasons (please see below comments). We did learn much during this study and will improve the methods in future developments, accordingly. Answer to 1): The focus is to improve the operationally running precipitation accumulation analyses, which use the spatial- and time resolution of 3 km and 1 hour, respectively. Gauge information is available as 1 hour accumulation from our real-time database. Therefore, the time resolution for analyzed accumulation is bound to be on hourly data. The verification within this article was performed during operational runs. Hence, rerunning longer periods would require resources not available due to all the extensive data input, which would have to be re-generate (including retrieval/extraction of data and format conversions). For the 4-days period (year 2014) we manually saved the input data, in order to rerun experiment where we exclude/include lighting from the data ingest and test different profile relationship generations. Answer to 2): Since the verification was performed during operational runs, the independent stations had to be set beforehand (i.e. excluded from the assimilation). Rerunning long periods with different independent stations, manually set for each event, would require extensive resources (see above explanation). By running the operational system for whole summer, we intended to retrieve a large statistical sample for verification. Unfortunately, summer 2015 was a period with very small amount of lightning cases. In the introduction we have added a short explanation of these limitations.

2. How are Fig 4 and Fig 8 obtained? Are they based on the dependent gauges? Do they show 1h estimates (I assume so since the figures show "mm/h" for the accumulations)? Using 1h estimates for the comparison with the dependent gauges (that are used on 1h scale for the RandB process) will have the Rad_LDA_RandB (1h product) necessarily being the best.

AUTHORS ANSWER: Note: After the revision of the paper, all figures and much of the text have been reorganized. Figure 4 was obtained from verification against dependent gauges and in Fig. 8 the independent gauges are used. This is now corrected and clarified in the new figure captions. Yes, they are both given as hourly accumulation values. We changed units to read "mm" and then, in text and figure captions, we mention that it is "hourly accumulation values". Table 1 and 2 shows results for the dependent gauges, here one can see that Rad_LDA_RandB give same results as the Rad_RandB. The same result is achieved from the independent gauges (now also mentioned in the article text).

3. I suggest to choose one between r2 and Pareson's correlation coefficient since the two statistics provide the same information. Moreover, basing results on RMSE can be tricky because errors are not weighted.

AUTHORS ANSWER: Yes, we agree. We have now removed the coefficient of determination (R2) from the verification. We would prefer to keep the RMSE, since it is widely used in literature and it is something that readers are used to interpret.

Minor comments:

1. The title should include more clearly the second objective of the study (impact of different integration time intervals in the radar-gauge correction method)

AUTHORS ANSWER: The title is now changed: "Improving the precipitation accumulation analysis using lightning measurements and different integration periods"

2. lines 1-5: the sentence is difficult to read. Moreover the second objective of the study should be better stressed. What about: "Two main objectives are addressed: (i) the assimilation of lightning observations in radar and gauge measurements and

(ii) the analysis of the impact of different integration time intervals in the radar-gauge correction method."

AUTHORS ANSWER: We agree and have now changed the first paragraph to read: "The focus of this article is to improve the precipitation accumulation analysis, with special focus on the intense precipitation events. Two main objectives are addressed: (i) the assimilation of lightning observations together with radar and gauge measurements and (ii) the analysis of the impact of different integration periods in the radar-gauge correction method."

3. line 6: is the reference Gregow et al. (2013)?

AUTHORS ANSWER: Yes, thank you. This is now corrected (year 2011 → 2013).

4. The state of art section (lines 28-39) is rather short and can be organized in a clearer way

AUTHORS ANSWER: We have added text and references to the Introduction and organized it to become more clear. Added text include paragraphs: "The research of combining radar and surface observations, to perform corrections to precipitation accumulation, is well explored. Many have made developments in this field and much literature is available, for example Sideris et al. (2014), Schiemann et al. (2011) and Goudenhoofdt and Delobbe (2009). Recently, Jewell and Gaussiat (2015) compared performances of different merging schemas, and noted a large difference between convective and stratiform situations. In their study, the non-parametric kriging with external drift (KEDn) outperformed other methods in accumulation period of 60 minutes. Wang et al (2015) developed a sophisticated method for urban hydrology, which preserves the non-normal charactersitics of the precipitation field. They also noticed that common methods have a tendency to smooth out the important but spatially limited extremes of precipitation." And: "Lightning is associated with convective precipitation, but in areas where a large portion of precipitation is stratiform, lightning data alone is not adequate for precipitation estimation. However, lightning has been used to complement

and improve other datasets. Morales and Agnastou (2003) combined lightning with satellite-based measurements to distinguish between convective and stratiform precipitation area and achieved a remarkable 31% bias reduction, compared to satellite-only techniques. Lightning has also been assimilated to numerical weather prediction models to improve the initialization process of the model. This can be done by blending them with other remote sensing data to create heating profiles (e.g. estimating the latent heat release when precipitation is condensed). Papadopulos et al. (2005) used lightning data to identify convective areas and then modified the model humidity profiles, allowing the model to produce convection and release latent heat using its own convective parameterization scheme. They combined lightning with 6-hourly gauge data, within a mesoscale model in the Mediterraiean area, and showed improvement in forecasts up to 12 hours lead time."

5. line 47: " usually with a higher quality than radar" a reference can be helpful

AUTHORS ANSWER: We have clarifed the sentence and added a reference in Sect. 2.1: "Rain gauges provide point observations of the accumulation. They are usually considered more accurate than radar, as point values, and are frequently used to correct the radar field (Wilson and Brandes, 1979)."

6. lines 54-55: "long" rather than "longer", "short" rather than "shorter"

AUTHORS ANSWER: This has now been changed.

7. line 62: more information about how "poor data quality" stations are identified is needed

AUTHORS ANSWER: Clarified in Sect. 2.1 by following sentence: "If measurements consistently indicate poor data quality, either manually identified from station error-logs or by inspecting the data, those stations are blacklisted within the LAPS process and do not contribute to the precipitation accumulation analysis."

8. line 70: Lat-Lon information are not shown in the figure

AUTHORS ANSWER: We have now rephrased the sentence to: "As Finland has no high mountains, the horizon of all the radars is near zero elevation with no major beam blockage, and, in general, the radar coverage is very good except in the most northern part of the country."

9. lines 70-72: something is missing in the sentence

AUTHORS ANSWER: Changed to read: "During year 2014 and 2015 the utilization rate was > 99%."

10. line 108: I couldn't find the work by Pessi and Albers, 2014

AUTHORS ANSWER: We have now updated the reference with a web-link of the presentation: https://ams.confex.com/ams/94Annual/webprogram/Paper238715.html

11. lines 120-124: this is not useful for the purposes of the paper

AUTHORS ANSWER: We would like to keep these sentences about existing "default profiles". Because, we believe it is relevant to point out that for experimental/operational usage, anywhere in the world, there is a direct possibilty to use and test the LDA method without collecting new, own statistical relationships.

12. line 121 and 127: I couldn't find the work by Pessi, 2013

AUTHORS ANSWER: We have now updated the reference with a web-link of the presentation: https://ams.confex.com/ams/93Annual/webprogram/Paper215562.html

13. line 184: why 0.3? more details are needed

AUTHORS ANSWER: The threshold value for the hourly surface gauge measurements, retrieved from FMI real-time database, is 0.254 mm/h (i.e. everything below 0.254 mm/h is just 0). The sentence changed and moved to Sect. 3.4: "In this study we apply a filter to the verification datasets, where hourly accumulation data less than 0.3 mm are discarded (due to the lowest threshold value of surface gauge measurements from FMI real-time database)."

14. Fig 7: the colors of the regression lines are not explained in the caption

AUTHORS ANSWER: We have clarifeid and added the following to figure caption "The corresponding regression lines are represented with same color as the markers, for each method."

Please also note the supplement to this comment:
http://www.hydrol-earth-syst-sci-discuss.net/hess-2016-113/hess-2016-113-AC1-supplement.pdf

---

## Author Comment (AC2) · 26 May 2016

H. Leijnse (Referee) hidde.leijnse@knmi.nl

Received and published: 17 April 2016

This paper describes an assessment of the quality of quantitative rainfall estimates using a combination radar, cloud-to-ground lightning, and rain gauge data. The effect of adding GC lightning data to radar data on rainfall accumulations is investigated, both before and after gauge adjustment. In these analyses, several methods of estimating relations between lightning activity and rain intensity are utilized. Furthermore, the effect of the length of the accumulation interval used for gauge adjustment is also

studied. The paper is interesting, and its topic relevant. It is not entirely clear to me what the main goal of this paper is. I think that the paper could benefit from a clearer description of what its main goals are, how the analyses that are presented contribute to these goals, and coming back to these goals in the Discussions and Conclusions section. When reading the paper for the first time I was sometimes confused because new analyses are proposed in the Results section and some of the methods described in the Methods section were not entirely clear to me. Hence, the paper could benefit from some restructuring, where all methods that are used are presented clearly in the Methods section. I think that the paper needs major revisions in order for it to be suitable for publication. More specific remarks are given below.

AUTHORS: The authors want to thank the reviewer for the professional and thorough revision of this paper. The paper has undergone a significant reorganization and has now a better structure. Please see the new updated article version, attached as Supplement.

Specific comments

1. Section 1, Given the fact that there are not very many lightning strikes in Finland, how much would you expect that adding this information would influence the final rainfall estimates? I think that this should be thoroughly discussed in the introduction of the paper.

AUTHORS ANSWER: Yes, it is correct that due to low lighting frequency in Finland the quantitative effect during a year is small. But the goal is to improve the quality of the few (but important) existing intense precipitation cases (i.e. causing flash floods etc). Here the LDA method have an impact, since the largest uncertainties took place during heavy rainfall (i.e. convective weather situations and lightning; Gregow et al., 2013). We have added text related to this in the Introduction section: "Radar reflectivity can in some cases suffer from poor quality, resulting from electronic mis-calibration, beam blocking, clutter, attenuation and overhanging precipitation (Saltikoff et al., 2010). In

some cases the radar can even be missing, due to upgrading or technical problems. Thunderstorms add probability of many of these problems in form of interruptions in electicity and telecommunications, and attenuation due to intervening heavy precipitation. In general, combining radar and rain gauge data is very difficult in the vicinity of heavy, local rain cells (Einfalt et al., 2005)." Also, the intention is to enlarge the analysis area to whole Scandinavia. For this reason the LDA will have a larger contribution to the precipitaion accumulation analysis, since there are gaps in radar coverage for this area and the retrieval of data is not always stable (i.e. radars can be missing more frequently from neighbouring countries). This is mentioned in the Introduction with following text: "Our situation is different from the above mentioned experiments because lightning activity is usually low in Finland, compared to warmer climates (Mäkelä et al., 2011). Also, our analysis area already has a good radar coverage and relatively evenly distributed network of 1 hour gauge measurements. However, if we want to enlarge the analysis area, we will soon go to either sea areas or neighbouring countries where availability of radar data and frequent gauge measurements is low. Our principal goal is to have as good analysis as possible, which is different from having a best analysis to start a model."

2. Section 3.3, It's not entirely clear how radar and lightning data are merged to come up with a final rainfall estimate. Did I understand correctly that the number of recorded GC lightning strikes within a LAPS pixel (3x3 km) and in a 5-minute interval are counted. These counts are then related to a vertical reflectivity profile, and subsequently the maximum of the radar reflectivity and the 'lightning reflectivity' are taken at a given height. The rainfall estimate is then based on the lowest data point. In practice, this means that the rainfall estimate is based on the maximum of the lowest-level radar reflectivity'. If this is indeed the case, the description of the method to estimate rain rates could be simplified and clarified. If not, I recommend clarifying this section.

AUTHORS ANSWER: Sections 3.2 and 3.3 are now merged. The text is reorganized

СЗ

and we have clarified the process better.

3. I.137-149, Given the fact that lightning only occurs in convective situations, it would make sense to me if a Z-R relation specifically derived for convective rain is used wherever lightning is observed. This would be a simple addition to the LDA that could improve results even further.

AUTHORS ANSWER: Thank you, this is a very good suggestion. This article presents the first results in an on-going process of developing the LAPS-LDA system at FMI. We have thought of different ways to improve the system, learned much during this study and the plan is to implement new routines, in future versions of LAPS-LDA system. The suggestion by reviewer is clearly one that should be considered then.

4. I.174-177, the rationale behind the regression part of the RandB method is that radar rainfall estimates often suffer from large-scale multiplicative biases, and that using regression on radar and gauge data can correct for this error. When adding lightning data to radar data, the errors are likely to be very different, and this could have a large effect on the final rainfall estimates. Something similar can be said for the Barnes-part of the RandB method, where the influence of a gauge correction is in general relatively large compared to the area affected by lightning. I therefore strongly suggest to add a discussion of this in the paper.

AUTHORS ANSWER: We agree that this is a simplification of mixing different scaleprocesses. In the Introduction we included following text: "Lightning is associated with convective precipitation, but in areas where a large portion of precipitation is stratiform, lightning data alone is not adequate for precipitation estimation. However, lightning has been used to complement and improve other datasets. Morales and Agnastou (2003) combined lightning with satellite-based measurements to distinguish between convective and stratiform precipitation area and achieved a remarkable 31% bias reduction, compared to satellite-only techniques. Lightning has also been assimilated to numerical weather prediction models to improve the initialization process of the model. This can be done by blending them with other remote sensing data to create heating profiles (e.g. estimating the latent heat release when precipitation is condensed). Papadopulos et al. (2005) used lightning data to identify convective areas and then modified the model humidity profiles, allowing the model to produce convection and release latent heat using its own convective parameterization scheme. They combined lightning with 6-hourly gauge data, within a mesoscale model in the Mediterraiean area, and showed improvement in forecasts up to 12 hours lead time." And we added text about this in the discussions section: "In the RandB-method the Regression is used to correct for large-scale multiplicative biases between radar and gauge data. In this article we introduce lightning into the RandB-method, as an additional data source. However, lightning errors are likely to be different from those of radar and gauges and this could have an effect on the methodology used here. In future developments, after collecting longer time series to quantify the nature of uncertainty of lightning-based precipitation estimates, we intend to improve the analysis in this direction."

5. I.176-177, What does it mean that Rad\_LDA\_Accum is the reference?

AUTHORS ANSWER: This sentence (now in Sect. 4.2) is changed to: "Note that Rad\_LDA\_Accum (e.g. a method not using RandB, as an reference) is included when comparing the results of different integration periods."

6. Section 4, Why are the graphs where rainfall intensities are compared plotted on log-log scales? If the aim is to study the performance of quantitative precipitation estimation algorithms for high intensities (as is stated in the paper), it would make most sense to me if these graphs were plotted using linear axes.

AUTHORS ANSWER: Yes, the intention is to increase the readability of high precipitation values but without disturbing the overal readability. Plotting the values on linear axes will decrease the readability of the low-middle values. The log-log scales was the best way we could produce these plots (according to us), after testing different plotting techniques (see below). Therefore we suggest to keep Figs. 5 and 7 with log-scales.

As an example we plot Fig. 5 in log-scale vs linear-scale (please see the attached Fig. 1). And it is the same with Fig. 4 and 8, the visualization of data is more clear with log-scales. Here we show Fig. 4a with log-scale vs linear-scale (please see the attached Fig. 2).

7. I.187-188, what exactly is meant by "the averaged (i.e. 50%-percentile) Rad-Lig reflectivity profiles from the LDA-method."? How were these profiles determined, and based on what data? I think this should be discussed in the Methods section.

AUTHORS ANSWER: It is now moved into Methods, Sect. 3.2 (merged and changed with other sections). The related text now reads: "For this study over Finland, climatological Rad-Lig reflectivity relationship profiles were estimated using NORDLIS-LLS lightning information and operational radar volume data from Finland area, during summer 2014. A total of approximately 220'000 lightning strokes were used for this calibration. The FMI-LAPS LDA is using 5 minutes interval of lightning- and radar data, within a LAPS grid-box of resolution 3\*3 km. The collected strokes are divided into binned categories using an exponential division (i.e. 2n...2n+1), following the same method used in Pessi (2013). This result in 6 different lightning categories (e.g. with 1, 2-3, 4-7, 8-15, 16-31 and 32-63 strokes) for the NORDLIS-LLS dataset. For each of these 6 categories, the average radar reflectivity profile is calculated and gives the Average Rad-Lig profiles (Fig. 3a), which is the baseline method. We extend this method to also calculate the 3'rd Quartile (i.e. 75%-percentile) and a Variable Quartile Rad-Lig profiles. The Variable Quartile method uses a range between 50%-percentile (for the lower dBZ values) up to the 95%-percentile (for the highest dBZ values)."

In this answer (to reviewer) we also provide a plot which visualize the process. For each category we collect the relevant radar reflectivity profiles. From these selections of profiles, the average is calculted and further used as the LDA-lightning profile (please see the attached Fig. 3).

8. I.192-200, I suggest to remove the R2 statistic, because it is simply the correlation

coefficient squared (see Eqs (6) and (7)) and it hence doesn't add any information relative to CORR.

AUTHORS ANSWER: We have now removed the R2 statistics from the text and tables.

9. I.202-207, The panels of Fig. 4 with LDA added (i.e. panels b and d) do not really add any information, as they are extremely similar to panels a and c, respectively. I therefore suggest making a remark in the text about this, and removing either panels a and c, or panels b and d.

AUTHORS ANSWER: We have added this into Discussion: "The accumulation products generated from RandB-method are corrected using gauge information. This process is influencing the final accumulation results much more than the contribution from the LDA-method (seen in Fig. 4 results from dependent dataset, where a, c and b, d panels, respectively, are almost identical). The same result was seen for the independent dataset." We suggest to keep Fig. 4 as it is. Removing either a,c- or b,d-panels, only mentioning this in the text, would most probably result in contradicting comments by other reviewers (i.e. that this should be shown with figures).

10. I.208-212, I would strongly suggest using different gauges for the independent measurements to test whether using LDA improves rain estimates, because this is what I understand the main objective of this paper to be.

AUTHORS ANSWER: The verification in this article was performed during operational LAPS runs (i.e. products are used within end-users applications). Seven independent stations were pre-selected (from different parts of Finland). Because of this we could not set more stations aside, without risking the quality of the end product. Re-running longer periods with different independent stations, manually set for each event and re-generate the extensive input datasets (retrieval/extraction of data, format conversions etc), would require resources not available. By running the operational system for whole summer, we intended to retrieve a large statistical sample for verification. Unfortunately, summer 2015 was a period with very small amount of lightning cases.

This restriction is now mentioned and explained in the Introduction: "The work reported here has been performed using the operational Local Analysis and Prediction System (LAPS), which is used in the wether service of Finnish Meteorological Institute (FMI). Testing new approaches in an operational system has its limitations in e.g. excluding independent reference stations. Also the possibilities to rerun cases with different settings have been limited. The benefit of the approach is that we can be sure that we only use data which is operationally available."

11. I.209-210, The use of a 25-day subset is introduced here. I suggest introducing this earlier in the paper (the Methods section). And if this subset is used, what is the added value of using the 4-day subset? I think the clarity of the paper would improve if either the 4-day or the 25-day subset is used.

AUTHORS ANSWER: The lightning information is local in terms of time (e.g. also in space). Therefore, the potential effects of the use of LDA is not detected when long periods (seasonal) are used for the assessment, as they are masked out. We are trying to show this by using different verification periods (summer-, 25- and down to the 4-days periods). The 4-days subset (for which we have saved all the extensive input data) also fills another purpose, namely being able to rerun and test different developments (such as the verification of average-, 3'rd- and Variable Quartile Rad-Lig profiles). The paper has undergone many changes and is reorganized. We now introduce the 25-day subset in Methods, Sect. 3.4, together with the other periods, as follows: "The verification periods consists of one long period ranging from 1 April to 1 September, 2015 (i.e. to avoid the winter season and snow precipitation). This dataset includes many precipitating cases without lightning and therefore, the effective impact by lightning is diluted (e.g. no influence by the LDA-method). Therefore, a subset of 25 days with frequent lightning (e.g. > 100 CG strokes/day) were selected from summer 2015. Additionally, in order to perform several autonomous experiments with the FMI-LAPS LDA system, a dataset consisting of four days with heavy rain and strong convection were used: 03, 23, 24 and 30 of July 2014 (hereafter 4-days period). These

were the 4 days with highest lightning intensity (e.g. > 100 strokes/day) in Finland, during year 2014."

12. I.225-238, It's unclear to me how the new profiles are exactly generated. I strongly suggest to include a good description of this in the Methods section (preferably in Section 3.2).

AUTHORS ANSWER: Please, also see reply to comment 7 here above. We have now moved and merged sections. The description of Rad-Lig relationship profiles is now better explained in Sect. 3.2.

13. I.240-245, Why not test sub-hourly scales?

AUTHORS ANSWER: The gauge information is available as 1 hour accumulation, from our FMI real-time database, and this is used in our operational runs. Therefore, the time resolution for analyzed accumulation is bound to be on hourly data.

Minor remarks

1. I.16, replace "such as;" by "such as" (remove semicolon)

AUTHORS ANSWER: This is done.

2. I.17, replace "eceonomically" by "economically"

AUTHORS ANSWER: This is done.

3. I.39, replace "leass" by "less"

AUTHORS ANSWER: This is done.

4. I.41, what is meant by "a timely accurate manner"?

AUTHORS ANSWER: This is changed to "...timely manner (i.e. near real-time).".

5. I.133, replace "resulting from;" by "resulting from" (remove semicolon)

AUTHORS ANSWER: This is done.

6. I.133-134, consider including clutter as an important source of error

AUTHORS ANSWER: Clutter has been added to the sentence.

7. I.144-145, do you mean to say here that convective rain is important for flooding events? I so, I suggest changing "such situations" to "convective events". The first time I read this sentence I interpreted "such situations" to be the drizzle that is mentioned in the previous sentence.

AUTHORS ANSWER: Yes, this is what we meant and it is now changed to "convective events".

8. I.187, the 50th percentile is not the average, but the median, and it is either the 50th percentile or the 50% quantile. So I suggest replacing "averaged (i.e. 50%- percentile)" by "median (i.e. 50% quantile)."

AUTHORS ANSWER: Correct, well spotted. We have remove the "(i.e. 50%-percentile)" here and in other places in the text, where this occur.

9. I.192, I suggest calling STDEV "relative standard deviation" or "logarithmic standard deviation" to make clear that it is different from a regular standard deviation.

AUTHORS ANSWER: We have now changed this to read "the logarithmic standard deviation".

Please also note the supplement to this comment: http://www.hydrol-earth-syst-sci-discuss.net/hess-2016-113/hess-2016-113-AC2supplement.pdf

Fig. 1. Refers to comment 6)

Fig. 2. Refers to comment 6)

Fig. 3. Refers to comment 7)

---

## Author Comment (AC3) · 26 May 2016

This paper addresses the question of how to the use of lightning data can help improve rainfall accumulation estimation. The topic is relevant and paper on this issue are welcomed. However the manuscript exhibit severe flaws, and cannot be published in its current form. The modification needed require in-depth modification.

AUTHORS: The authors want to thank the reviewer for the professional and thorough revision of this paper. The new updated article version is attached as Supplement (see

PDF-file).

General comments:

Overall the paper is quite difficult to read, many processing techniques are tested on various data sets, presented not at the same time. Maybe a scheme summarizing the techniques would help. I think the paper should also be organized better. A solution could be to present "data", then methods with a subsection on the various products from the radar, gauge and lightning, and a subsection on how the comparison is performed.

AUTHORS ANSWER: The paper has undergone a significant reorganization, which has now improved the readability accordingly. The methods have now been bundled from different sections, the observations have a better structure and the result section is more concise.

There seems to be a contradiction between the abstract and the content. abstract l.8-9 : "the performed... usefullness of..." and l.201 "The overall result... neutral to positive impact..." and same comment on the dependent sub-set (l.211-212).

AUTHORS ANSWER: We have changed the second paragraph in abstract to be more consistent with the result section, which has been rewritten: "Lightning data does improve the analysis when no radar is available, and even with radar, lightnings have a neutral to positive impact on the results."

The conclusion seems to be that basically when radar data is available lightning data is rather useless. This is already a result that should be stressed (it is already mentioned). That said I have the feeling that the paper could be more interesting by shifting its scope to how to estimate rainfall (locally) from lightning data when no radar data is available (this would correspond to developing more in depth what is done with figure 7). After this analysis, you could practically test the interest by artificially removing some radar data.

AUTHORS ANSWER: We now stress the importance of lightning data for situations of no radar information in the abstract, the result- and discussion sections. The results in this article was performed during operational LAPS runs, i.e. with all input data availabe at that time. Hence, rerunning longer periods would require resources not available, due to all the extensive data input needed (i.e. regenerate the data input and format conversions). For the 4-days period we manually saved the input data in order to rerun experiment, where we exclude/include lighting from the data ingest and, additionally, test different profile relationship generations. This is now explained in the introduction: "The work reported here has been performed using the operational Local Analysis and Prediction System (LAPS), which is used in the wether service of Finnish Meteorological Institute (FMI). Testing new approaches in an operational system has its limitations in e.g. excluding independent reference stations. Also the possibilities to rerun cases with different settings have been limited. The benefit of the approach is that we can be sure that we only use data which is operationally available."

Detailed comments:

The title "radar -,gauge-" formula is not very clear.

AUTHORS ANSWER: The title is changed to: "Improving the precipitation accumulation analysis using lightning measurements and different integration periods"

1) Introduction - It should be extended to include a state of art section on the actual topic of the paper, i.e. lightning measurement assimilation for rainfall or more generally in meteorology. (ex among many others: Papadopoulos et al. 2005; Morales et al 2003)

AUTHORS ANSWER: We have now included text in the Introduction section, where work on this topic is elaborated (including references) as follows: "Lightning is associated with convective precipitation, but in areas where a large portion of precipitation is stratiform, lightning data alone is not adequate for precipitation estimation. However, lightning has been used to complement and improve other datasets. Morales

and Agnastou (2003) combined lightning with satellite-based measurements to distinguish between convective and stratiform precipitation area and achieved a remarkable 31% bias reduction, compared to satellite-only techniques. Lightning has also been assimilated to numerical weather prediction models to improve the initialization process of the model. This can be done by blending them with other remote sensing data to create heating profiles (e.g. estimating the latent heat release when precipitation is condensed). Papadopulos et al. (2005) used lightning data to identify convective areas and then modified the model humidity profiles, allowing the model to produce convection and release latent heat using its own convective parameterization scheme. They combined lightning with 6-hourly gauge data, within a mesoscale model in the Mediterraiean area, and showed improvement in forecasts up to 12 hours lead time. Our situation is different from the above mentioned experiments because lightning activity is usually low in Finland, compared to warmer climates (Mäkelä et al., 2011). Also, our analysis area already has a good radar coverage and relatively evenly distributed network of 1 hour gauge measurements. However, if we want to enlarge the analysis area, we will soon go to either sea areas or neighbouring countries where availability of radar data and frequent gauge measurements is low. Our principal goal is to have as good analysis as possible, which is different from having a best analysis to start a model."

2) Observations and instrumentation - p2 l.49 : "LAPS" is used but was not defined before in the manuscript

AUTHORS ANSWER: LAPS is now defined properly.

- p3 l.76: "as as proxy", one "as" should be removed

AUTHORS ANSWER: This is corrected.

- section 2.2 : how mosaicking is done ? Some more detail on the radar processing should be provided. How dual polarisation is used? The differences in terms of sampling area between rain gauges and radar are almost not mentioned (l.81). This

discussion should at least be expanded because it can have an influence on the standard scores used after. See for example reference such as Jaffrain and Berne 2012; Gires et al. 2014; on this issue

AUTHORS ANSWER: The text related to LAPS processes is reorganized (now in Sect. 3.1), including a reference to (Albers et al., 1996): "The Finnish radar volume scans are read into LAPS as NetCDF format files, thereafter the data is remapped to LAPS internal Cartesian grid and the mosaic process combines data of the different radar stations (Albers et al., 1996)."

A sentence related to the use of dual polarisation radar is included (Sect. 2.2): "At the moment, the quantitative precipitation estimation based on dual-polarization is not used operationally in FMI, but the polarimetric properties contribute to the improved clutter cancellation (i.e. removal of non-meteorological echoes, especially sea clutter, birds and insects)."

The differences in sampling size (both spacial and temporal) has now been inlcuded in the introduction section, with text and references as follows: "Comparing radars and gauges, an additional challenge arises from the different sampling sizes of the instruments. Radar measurement volume can be several kilometers wide and thick (one degree beam is approximately 5 kilometres wide at 250 kilometres), while the measurement area of a gauge is 400 cm2 (weighing gauges) or 100 cm3 (optical instruments). Part of the disparateness of radar and gauge measurements is due to variability of the raindrop size distribution within area of a single radar pixel. Jaffrain and Berne (2012) have observed variability up to 15% of the rainrate in a 1x1 km pixel, with timesteps of 1 minute. Gires et al (2014) have shown that the scale difference has an effect in verification measures (such as normalized bias, e.g. RMSE) but it decreases with growing accumulation time (e.g. from 5 to 60 minutes). In our study, the 60 minutes accumulation period is smoothing some of the differences."

3) Methods

- l94 : randB method mentioned but not defined after

AUTHORS ANSWER: This has been corrected.

- section 3.1 : more details on LAPS are needed. Sentences such as "LAPS uses statistical methods to perform high-resolution analysis" are too general for a scientific paper. It is not clear what is the purpose of LAPS and what data is used out of it, and how it is related to the radar mosaic product of FMI.

AUTHORS ANSWER: We have now added more information and references to the LAPS processes: "The FMI-LAPS use a pressure coordinate system including 44 vertical levels distributed with a higher resolution (e.g. 10 hPa) at lower altitudes and decreasing with height. The horizontal resolution is 3 kilometres and the temporal resolution is 1 hour. The domain used in this article covers the whole Finland and some parts of the neighbouring countries (Fig. 1b). LAPS highly relies on the existence of high-resolution observational network, in both space and time, and especially on remote sensing data. The FMI-LAPS is able to process several types of in-situ and remotely sensed observations (Koskinen et al., 2011), among which radar reflectivity, weighting gauges and road weather observations are used for calculating the precipitation accumulation. The Finnish radar volume scans are read into LAPS as NetCDF format files, thereafter the data is remapped to LAPS internal Cartesian grid and the mosaic process combines data of the different radar stations (Albers et al., 1996). The rain-rates are calculated from the lowest levels of the LAPS 3D radar mosaic data, via the standard Z-R formula (Marshall and Palmer, 1948), which is then used for precipitation accumulation calculations (see Sect. 3.2). Other information on observational usage, first-guess fields, the coordinate system etc. is described in Gregow et al. (2013)."

- Section 3.2: Some indication on the number of lightning strokes used to calibrate the relation should be given. Figure 3.b : the vertical scale should be changed to improve the visibility of the relation. Could you confirm that the temporal resolution is indeed of

5 min

AUTHORS ANSWER: We included (in Sect. 3.2): "A total of approximately 220'000 lightning strokes were used for this calibration." We have changed y-axis scale in Fig. 3b, to make improve the readability. Yes, the radar and lightning data have 5 minutes resolution and the final accumulation product is hourly.

- Eq 3 : so the dual pol capacities are not used ? AUTHORS ANSWER: At the moment, the quantitative precipitation estimation based on dual-polarization is not used, but the polarimetric properties contribute to the improved clutter cancellation (see also answer above).

- l.151-152 : when merging radar and lighting data, why choosing the maximum ? When radar data is available, is not radar more reliable than one choice among the 6 different profiles for lightning ? Please justify this choice.

AUTHORS ANSWER: In most cases the radar data are more reliable, compared to lightning-profiles. Though, in cases where there occur attenuation or the grid-point is at the far end of radar coverage, the lightning-profiles could be of better quality and hence, in these situtaions would have higher reflectivity values (therefore we choose the dataset with higher value). We included text explaining this: "The logic behind this is that the radars are more likely to underestimate, than overestimate the precipitation (due to attenuation, beam blocking or the nearest radar missing from network; e.g. Battan, 1973 and Germann, 1999), especially in thunderstorm situations."

- Section 3.4 : why only one sub section? You have to say at least few words on Rand B method and Barnes analysis. It is very difficult to understand this section with so little explanations.

AUTHORS ANSWER: We admit we were perhaps too careful not to repeat too much from the earlier paper in Sect. 3.4. We now added new text describing the RandB-method: "The FMI-LAPS RandB-method corrects the precipitation accumulation estimates using radar and gauges datasets. The first step in this method is to make the radar-gauge correction using the Regression method. Data of hourly accumulation values are derived from the gauge-radar pairs within the LAPS grid (i.e. from same location and time), and from this a linear regression function can be established. The corrections from Regression method is applied to the whole radar accumulation field and thereafter used as input for the second step, the Barnes analysis. Within LAPS routines the Barnes interpolation converge the radar field towards gauge accumulation measurements at smaller areas (i.e. for gauge station surroundings). Several iterative correction steps are performed within the Barnes analysis, adjusting the final accumuation. The FMI-LAPS RandB-method is described in more details in Gregow et al. (2013)."

4) Results and verification

- p. 7 is methodology and should be moved in the corresponding section. Please confirm that in eq. 4-7, values are taken at the hourly resolution ? It might be interesting to test other time steps.

AUTHORS ANSWER: The result section has been rewritten and parts related to methods have been moved accordingly. Yes, we confirm that these values are hourly. We agree, it would be interesting and important to use lower time-resolution. However, the surface gauge measurements, coming from FMI real-time database, are given as hourly values. Therefore we are restricted to make the corrections using the time resolution of 1 hour.

- l.187-188 : "the avg Radlig reflectivity profile. Please clarify ?

AUTHORS ANSWER: This text is moved and rewritten in methods section (Sect. 3.2, now merged with Sect. 3.3).

- Figure 4 : please clarify what is plotted and how it was obtained. It is also almost not commented in the text.

AUTHORS ANSWER: The figure caption has been updated with more information: "Figure 4. The FMI-LAPS precipitation accumulation (described in plots with density iso-lines of hourly accumulation values, in mm and log-scale) calculated using 4 different methods: a) Rad_Accum, b) Rad_LDA_Accum, c) Rad_RandB and in d) Rad_LDA_RandB. Results are from the dependent gauge dataset during summer 2015. Shown is also the best fit line (1:1)."

- Figure 5 : again mention time steps used (1h ?). l.220-224 : the figure should be more commented. - l.228-230 : the quantiles mentioned are not clear.

AUTHORS ANSWER: We added information about hourly accumulation into the caption of Fig. 5. The result section has been rewritten to become more readable and asserted.

- Figure 7 and comments : the change in quantile seems to improve rainfall estimates. Why was not it used in the first place ?

AUTHORS ANSWER: Note: Figs. 6-8 have now changed order. Why not use the Quartile method in first place? This is because we had a test-dataset, a 4-days case from year 2014 (with all the extensive input data), which was used to perform autonomous experiments/reruns. But, the idea of calculating/testing different Quartile profiles were not thought of until we saw the results from summer 2015. Then, we made reruns using data from 2014 to establish these findings (i.e. Variable Quartile approach give the best profiles).

- Figure 8 : please comment more the figure...

AUTHORS ANSWER: We have added text to the figure caption.

References (reviewer):

Extending the Capabilities of High-Frequency Rainfall Estimation from Geostationary-Based Satellite Infrared via a Network of Long-Range Lightning Observations Carlos A. Morales and Emmanouil N. Anagnostou Journal of Hydrometeorology 2003 4:2,

141-159

Improving Convective Precipitation Forecasting through Assimilation of Regional Lightning Measurements in a Mesoscale Model Anastasios Papadopoulos, Themis G. Chronis, and Emmanouil N. Anagnostou Monthly Weather Review 2005 133:7, 1961-1977

Gires, A. et al., 2014. Influence of small scale rainfall variability on standard comparison tools between radar and rain gauge data. Atmospheric Research, 138(0): 125-138.

Jaffrain, J. and Berne, A., 2012. Influence of the Subgrid Variability of the Raindrop Size Distribution on Radar Rainfall Estimators. Journal of Applied Meteorology and Climatology, 51(4), 780-785.

Please also note the supplement to this comment:
http://www.hydrol-earth-syst-sci-discuss.net/hess-2016-113/hess-2016-113-AC3-supplement.pdf

**Supplement:**

**Improving the precipitation accumulation analysis using lightning measurements and different integration periods**

E. Gregow[1], A. Pessi[2], A. Mäkelä[1] and E. Saltikoff[1]

5 [1]Finnish Meteorological Institute, P.O. Box 503, FIN-00101 Helsinki, Finland

[2]Vaisala, 3 Lan Dr., Westford, MA 01886, USA

*Correspondence to:* E. Gregow (erik.gregow@fmi.fi)

**Abstract.** The focus of this article is to improve the precipitation accumulation analysis, with
10 special focus on the intense precipitation events. Two main objectives are addressed: (i) the assimilation of lightning observations together with radar and gauge measurements and (ii) the analysis of the impact of different integration periods in the radar-gauge correction method. The article is a continuation of previous work in the same research-field, by Gregow et al. (2013).

15 A new Lightning Data Assimilation (LDA) method has been implemented and validated within the Finnish Meteorological Institute (FMI) - Local Analysis and Prediction System (LAPS). Lightning data does improve the analysis when no radar is available, and even with radar, lightnings have a neutral to 
[revised manuscript text omitted]

335 A slight improvement in the accumulation analysis, using lightning information, can be seen in the RMSE from the 25 days dataset of frequent thunderstorms (Table 1; compare Rad_Accum and Rad_LDA_Accum). The effect is only visible when comparing with the dependent gauges, as no thunderstorms occurred at the seven independent stations. When the verification period is extended to the entire summer of 2015 (i.e. including days with no

340 thunderstorms) the independent stations show some improvement (Table 2). The correlation (i.e. CORR) is higher for the Rad_LDA_Accum independent data (compared to Rad_Accum), and even though the RMSE is higher, the STDEV has been improved. The overall impact for the dependent stations is neutral (Table 2 and Fig. 4; compare Rad_Accum and Rad_LDA_Accum). For both the 25 days and whole summer period the Rad_RandB and

345 Rad_LDA_RandB give the best results, with similar scores.

Comparing the accumulation results from the 4-days period for radar alone (i.e. Rad_Accum; black markers in Fig. 5) and lightning alone (i.e. LDA_Accum; red markers in Fig. 5), it is clear that the use of LDA_Accum is less accurate than Radar_Accum results. Figure 5 also show that the Rad_LDA_Accum estimates (using the baseline method, with Average Rad-Lig

350 profiles) are amplified over the whole range of precipitation values, compared to Rad_Accum (Fig. 5; compare the blue with the black markers). For the high accumulation values (> 5 mm/h) this is a positive effect, while in lower range (< 5 mm/h) there is an overestimation of the results. Note that the plot uses log-scale at each axis.

Figure 6 show the results using different Rad-Lig profiles, i.e. Average-, 3'rd Quartile- and

355 Variable Quartile profiles. The results are validated against Rad_Accum. The precipitation accumulation estimates are improved at high accumulation values (> 5 mm), using either 3'rd- or Variable Quartile profiles. Simultaneously, they both does add to the overestimate in low accumulation values (< 5 mm). Note the use of log-scale, which enlarges the differences in the range of low values and reduces it in high ranges. The 3'rd Quartile profiles gives the

360 largest overestimate, over the whole accumulation scale.

**4.2 RandB-method and impact from different integration periods**

The plotted results of different time sampling periods are seen in Fig. 7, with verification against the independent stations. The Rad_LDA_RandB (i.e. using observations from the latest 1 hour) does give the best result, when compared to Rad_LDA_Accum, Rad_LDA_RandB, Rad_LDA_RandB_6hr, Rad_LDA_RandB_12hr, Rad_LDA_RandB_24hr and the Rad_LDA_RandB_7d output. The statistical scores shown in Table 3 also imply the same result. Note that Rad_LDA_Accum (e.g. a method not using RandB, as an reference) is included when comparing the results of different integration periods.

**Discussions and conclusions**

The aim of this article is to describe new methods on how to improve the hourly precipitation accumulation estimates, especially for heavy rainfall events (> 5 mm) and, if possible, also the low-valued ranges (< 5 mm) or at least leave them as unaffected as possible.

The strength of the LDA-method is that the radar and lightning information can be merged and complement each other. This is especially important in areas of poor, or even none, radar coverage, where the lightning information will improve the hourly precipitation accumulation analysis. The results in this article are limited to Finland but considering extending this area to include Scandinavia, the LDA-method will become even more useful. The long verification period (i.e. summer 2015) had fewer days of lighting compared to other years (on average) and therefore, the verification dataset was limited. Nevertheless, the summer period and the subset of 25 lighting instense days show neutral to positive impact in the reuslts, using the LDA-method (Table 1-2 and Fig. 4). 
[revised manuscript text omitted]

[Figure]

Figure 8. Example of lowest level reflectivity field from a) radar alone and b) converted
585 lightning locator analysis alone (via LDA system) for 30 July 2014 at 16 UTC. Reflectivity
color scale is shown below plots

---

## Author Comment (AC4) · 26 May 2016

S. Ochoa Rodriguez (Referee) s.ochoa-rodriguez@imperial.ac.uk

This paper explores the use of cloud-to-ground lightening data for improving radar-based (and raingauge-adjusted radar-based) precipitation estimation, with a focus on high intensity, convective storms. The study includes an evaluation of resulting rainfall estimates at different temporal accumulations.

The topic of the paper is interesting and the results are potentially useful. However, the paper has a number of major flaws which should be addressed before it can be

published.

AUTHORS: The authors want to thank the reviewer for the professional and thorough revision of this paper. The new updated article version is attached as Supplement (see PDF-file).

General comments / major flaws:

- The structure of the article is rather unorganised and the description of data, methods and results are often unclear. The description of the way in which lightening and radar data are merged is very unclear (and bits and pieces of the methodology are spread throughout several sections of the paper), descriptions of the methods are included in the data and results sections, amongst other things (more details are provided below).

AUTHORS ANSWER: Also other reviewers have commented on this and therefore the article has now been reorganized in several ways: - Methodology is now better collected into Sect. 3 (moved text from Sect. 4, merged 3.2 with 3.3 etc). - Lightning and radar merging is now better explained. - Result section is now more asserted and concise. - Plus many other changes (see below comments). We believe this has improved the readablilty and objectives of the paper.

- The fact that only 7 independent rain gauges are used for evaluation renders the results statistically weak. This issue is so critical that in one of the cases (Table 2), there are simply no data available for evaluation from any of the independent gauges. Why out of 472 available rain gauges would you only select 7 for independent evaluation? I am aware that you also present the results of performance statistics at nonindependent rain gauge locations; however, I truly believe that more interesting results could be achieved if either more independent rain gauges were used or if a crossvalidation approach were implemented.

AUTHORS ANSWER: The results in this article were performed during operational LAPS runs, we could not set more stations aside, without risking the quality of the end-

users products and applications. This is now mentioned in the introduction section: "The work reported here has been performed using the operational Local Analysis and Prediction System (LAPS), which is used in the wether service of Finnish Meteorological Institute (FMI). Testing new approaches in an operational system has its limitations in e.g. excluding independent reference stations. Also the possibilities to rerun cases with different settings have been limited. The benefit of the approach is that we can be sure that we only use data which is operationally available." Unfortunately the summer of 2015 had an unusually low frequency of lightning, which limited the statistics for this study.

- The results, as currently presented, are too vague and far from the objectives initially set in the abstract and introduction. As stated in the title of the paper, you aim at "improving precipitation accumulation analysis using radar, gauge and lightening measurements". Throughout the paper, the focus is mostly on the added value of lightening data, which does not really lead to significant improvements in radar-based QPEs, let alone gauge-based adjusted radar QPEs. One option would be to change the focus (and consequently the title) of the paper to "the added value of lightening measurements" for generation of QPEs. The added value could be more clearly assessed under different scenarios, including with and without radar data available, with / without rain gauge data. From the results you present, the real advantage of lightening data appears to be in cases when radar data are not available (which can occur either because there simply are no weather radars, due to malfunctioning of the radar or to issues such as beam blockage and/or attenuation); this is very valuable and should be made clearer (and, from my point of view, justifies changing the focus of the paper/ changing the way in which objectives and results are described).

AUTHORS ANSWER: We now stress the importance of using lightning data (especially in case of no radar data availability), in the abstract, result and discsussion sections. Also, the title is changed: "Improving the precipitation accumulation analysis using lightning measurements and different integration periods"

Detailed comments:

Introduction: Work by the co-authors of the paper is often cited and a proper review of relevant literature on the actual topic of the paper is lacking. The introduction should be extended to include: - A review on the use of lightening data for QPE generation

AUTHORS ANSWER: We have now included text in the introduction section, where work on this topic is elaborated (including references) as follows: "Lightning is associated with convective precipitation, but in areas where a large portion of precipitation is stratiform, lightning data alone is not adequate for precipitation estimation. However, lightning has been used to complement and improve other datasets. Morales and Agnastou (2003) combined lightning with satellite-based measurements to distinguish between convective and stratiform precipitation area and achieved a remarkable 31% bias reduction, compared to satellite-only techniques. Lightning has also been assimilated to numerical weather prediction models to improve the initialization process of the model. This can be done by blending them with other remote sensing data to create heating profiles (e.g. estimating the latent heat release when precipitation is condensed). Papadopulos et al. (2005) used lightning data to identify convective areas and then modified the model humidity profiles, allowing the model to produce convection and release latent heat using its own convective parameterization scheme. They combined lightning with 6-hourly gauge data, within a mesoscale model in the Mediterraiean area, and showed improvement in forecasts up to 12 hours lead time. Our situation is different from the above mentioned experiments because lightning activity is usually low in Finland, compared to warmer climates (Mäkelä et al., 2011). Also, our analysis area already has a good radar coverage and relatively evenly distributed network of 1 hour gauge measurements. However, if we want to enlarge the analysis area, we will soon go to either sea areas or neighbouring countries where availability of radar data and frequent gauge measurements is low. Our principal goal is to have as good analysis as possible, which is different from having a best analysis to start a model."

- A more in depth and critical discussion of adjustment of radar QPEs. The only comment so far is that "the use of monthly adjustment factors leads to less than optimal results". Several studies have been carried out which have shown a clear advantage of adjusting radar QPEs based on rain gauge data and other sources of information, with corrections implemented at significantly shorter time scales (as compared to the monthly one that you mention). Also, since the focus of your study is mostly on convective storms, it may be worth discussing the performance of merging techniques for convective storms, which is still a problematic issue (see Jewell & Gaussiat, 2015; Wang et al., 2015) and which could be a case in which lightening information could be useful.

AUTHORS ANSWER: Together with previous answer (see above) we have also added the following text into the introduction part: "The research of combining radar and surface observations, to perform corrections to precipitation accumulation, is well explored. Many have made developments in this field and much literature is available, for example Sideris et al. (2014), Schiemann et al. (2011) and Goudenhoofdt and Delobbe (2009). Recently, Jewell and Gaussiat (2015) compared performances of different merging schemas, and noted a large difference between convective and stratiform situations. In their study, the non-parametric kriging with external drift (KEDn) outperformed other methods in accumulation period of 60 minutes. Wang et al (2015) developed a sophisticated method for urban hydrology, which preserves the non-normal charactersitics of the precipitation field. They also noticed that common methods have a tendency to smooth out the important but spatially limited extremes of precipitation."

- A review and discussion on the topic of the impact of temporal aggregation on precipitation products. The impact of the temporal scale at which adjustments are performed is a key part of this study. This issue has been discussed in a number of papers which should be reviewed and in the light of which the results of the present study should be analysed (e.g. you should discuss how the optimal temporal resolution that you found (1h) compares to that found in previous studies for the case of convective storms). See

for example Berndt et al. 2014.

AUTHORS ANSWER: This is addressed by including following section in the introduction: "Comparing radars and gauges, an additional challenge arises from the different sampling sizes of the instruments. Radar measurement volume can be several kilometers wide and thick (one degree beam is approximately 5 kilometres wide at 250 kilometres), while the measurement area of a gauge is 400 cm2 (weighing gauges) or 100 cm3 (optical instruments). Part of the disparateness of radar and gauge measurements is due to variability of the raindrop size distribution within area of a single radar pixel. Jaffrain and Berne (2012) have observed variability up to 15% of the rainrate in a 1x1 km pixel, with timesteps of 1 minute. Gires et al (2014) have shown that the scale difference has an effect in verification measures (such as normalized bias, e.g. RMSE) but it decreases with growing accumulation time (e.g. from 5 to 60 minutes). In our study, the 60 minutes accumulation period is smoothing some of the differences."

- I would suggest to remove irrelevant information, such as L26-27 (projected annual precipitation in Northern EU) and keep the introduction focused on the topic of the paper.

AUTHORS ANSWER: This sentence has now been removed.

Data Section: - L47-50 include description of methods and should be removed from the data section. I would suggest that you simply say that three data sources are employed in this study and then go on to explain them (without starting to describe the LAPS, which should be done in the methods section).

AUTHORS ANSWER: Text have been modified (text about rain gauges moved into Sect. 2.1) and now follows your suggestion (Sect. 2): "Here we describe the three data sources employed in this study: rain gauge observations, radar and lightning observations."

- Section 2.1 – I would change title to "Ground rain gauges" or would at least include

the word "rain gauge" in it.

AUTHORS ANSWER: Title has been changed to "Rain gauge observations"

- Section 2.1, L57-58 is a repetition of L52; try merging these sentences.

AUTHORS ANSWER: The sentences have been merged and the whole section reorganized.

- Section 2.3: A bit more details about lightening sensors would be desirable. It would help the reader understand how is it that lightening measurements can be translated into vertical 'radar' rainfall profiles and so on. I am aware that this information can be found in other papers, but I think it would be helpful and interesting for the reader to find a brief description of the sensors here. Just in the same way that you provide a brief overview of radar QPE generation.

AUTHORS ANSWER: We have now added more details about the lightning sensors: "Lightning location sensors detect the electromagnetic (EM) signals emitted by lightning return strokes, measure the signal azimuth and exact time (GPS). Sensors send these information to the central processing computer in real time which combines them, optimises the most probable strike point and outputs this information to the end user. More detailed informationÂăof LLS principles are described in Cummins et al. (1998)." Also, the section related to translating radar-lightning into profiles has been rewritten and made more clear.

- While all but one of the Finnish radars are dual-pol, it appears that dual-pol parameters are not being used for QPE generation. The authors mention that a single Z-R relationship is used, which implies simplification and assumptions about variable dropsize distribution and the like. Such simplification could clearly be avoided were dual-pol parameters used. It should be made clear (in Section 2.2) that dual-pol parameters are not being used at all and the implications of this should be discussed (I reckon that the use of dual-pol parameters would lead to much larger improvements in the quality of

radar QPEs than the use lightening information).

AUTHORS ANSWER: Thank you, we agree. Due to cold climate and relatively long distances between radars, a significant part of our radar measurements are made above melting layer, so the benefits of pure KDP-based QPE are not as obvious as in some other areas. We have tested the use of PhiDP-based attenuation correction methods, but some uncertainties remains. We have now made this clearer by adding following sentence into Sect. 2.2: "At the moment, the quantitative precipitation estimation based on dual-polarization is not used operationally in FMI, but the polarimetric properties contribute to the improved clutter cancellation (i.e. removal of non-meteorological echoes, especially sea clutter, birds and insects)."

Methods Section: this section is rather unorganised and a thorough re-structuring would be desirable.

AUTHORS ANSWER: We have made a thorough reorganization of the methods sections, following the suggestions by reviewer (see comments below). Note: Also the other reviewers comments have been implemented and the final structure has now become better.

Some specific comments/suggestions are the following: - Section 3.1, L97: ": : : where a dense observational input, from several sources, are fitted to a coarser background model first-guess field". Please indicate which data sources are used.

AUTHORS ANSWER: This section has now been enlarged and we inlcude a sentence and reference related to this: "The FMI-LAPS is able to process several types of in-situ and remotely sensed observations (Koskinen et al., 2011), among which radar reflectivity, weighting gauges and road weather observations are used for calculating the precipitation accumulation." We prefer not to mention all observational input, because most of them are not relevant for the accumulation process (in previous article, i.e. Gregow et al., 2013, this was pointed out by reviewers and we had to remove this information). But for your information, FMI-LAPS is using input from satellite, radar,

lightning, synop, metar, local surface observational network, soundings, air-plane reports, lidars and background model.

- Section 3.1, L102: you indicate the spatial resolution of the FMI-LAPS output, but not the temporal resolution. From subsequent sections I gather that the temporal resolution is 5 min, but it should be clearly indicated in Section 3.1.

AUTHORS ANSWER: FMI-LAPS temporal resolution is 1 hour and this has been added (together with more information): "The FMI-LAPS use a pressure coordinate system including 44 vertical levels distributed with a higher resolution (e.g. 10 hPa) at lower altitudes and decreasing with height. The horizontal resolution is 3 kilometres and the temporal resolution is 1 hour." Note: FMI-LAPS produce hourly analysis, i.e. the accumulation product is for 1 hour and calculated using 5 minutes segments of radar and lightning data. Correction with rain gauges are done using the hourly data (since rain gauge data is availabe as 1 hour accumulation from our FMI real-time database).

- Sections 3.2 and 3.3: these two sections present overlapping information and a clear and integrated description of the integration of lightening data with radar data is missing. I suggest merging these two sections and producing a new and clearer description of the merging method.

AUTHORS ANSWER: The sections 3.2 and 3.3 have now been merged, according to suggestion, and the new section has become more clear and readable.

Results section: - Why using coefficient of determination AND Pearson's correlation coefficient? Their only difference is that the correlation coefficient has a sign, so it may be useful to include both in cases where you expect the correlation coefficient to take negative values. Since this is not the case here, the two performance measures provide very similar information. I would suggest to use only one of the two.

AUTHORS ANSWER: Yes, we agree. We have now removed the coefficient of determination (R2) from the verification.

- A description of the log STDEV is missing and a justification for using a log-STDEV instead of the non-log STDEV should be included.

AUTHORS ANSWER: An explenation to STDEV has now been included: "STDEV quantifies the amount of variation (i.e. spread) of a dataset. A low STDEV indicates that the data points tend to be close to the mean value of the dataset. Here we use the logarithm of the quotients, in order to get the datasets closer to be normally distributed."

- As mentioned above, this section (results) is unorganised and includes a great deal of description of methods, which should be transferred to Section 3. Also, results should be presented in a more concise and assertive manner (comments such as "neutral to positive impact" should be removed). As suggested above, a shift in the focus of the paper could make it easier to describe the results in a more assertive / critical manner.

AUTHORS ANSWER: The results are now presented in a more clear and assertive way. Text related to methods have been moved accordingly, part of text is moved to discsussion and unnecessary subjective comments have been removed.

- A scale bar should be included in Figure 1. AUTHORS ANSWER: We have now included a scale bar in Fig. 1. (Scale bar in Fig. 2 was also corrected)

- Why using log scales in figures 4, 5, 7 and 8? I think a normal (linear) scale would be better. A linear scale is normally used in papers on this topic, so readers are used to it. I do not see any added value in using a log scale and I do think that it hinders interpretation of results.

AUTHORS ANSWER: The intention is to increase the readability of high precipitation values but without disturbring the overal readability. Plotting the values on linear axes will decrease the overal readability, which has been seen after testing different plotting techniques. The log-log scales was the best way to produce the plots (according to us). Therefore we prefer to keep Fig. 5 and 7 with log-scales. As an example we here

plot log-scale vs linear-scale (please see Fig. 1 attached below).

It is the same with Fig. 4 and 8, the visualization of the results is more clear and better with log-scales. Here we show Fig. 4a with log-scale vs linear-scale (please see Fig. 2 attached below).

- I suggest using mm to indicate rainfall accumulations (accompanied by the temporal aggregation scale), instead of using mm/h (which is the unit normally used for intensities).

AUTHORS ANSWER: We have now changed the units to "mm" and explain that this is hourly accumulation values in the figure captions (Fig. 4, 5, 7 and 8).

Other comments: - Why not work with sub-hourly temporal accumulations? The lifetime of convective cells is often < 1 h. Since the focus of the study is on convective storms and lightening and radar data are available at high temporal resolution (5 min), it would make sense to evaluate sub-hourly scales.

AUTHORS ANSWER: The limiting factor for this is the gauge information, which is available as 1 hour accumulation from FMI real-time database. Therefore, the time resolution for analysis is bound to be as hourly accumulation data.

- Misuse of semicolons throughout the paper. The semicolon should be either removed or replaced by a colon. E.g.: L16: "such as;" (remove semicolon) L133: "resulting from;" (either remove or change to colon) L218: "analysis time;" (change to colon) Many others! Please check.

AUTHORS ANSWER: We have now made corrections to the above comments (plus several other places in the text).

REFERENCES (by reviewer):

Berndt, C., Rabiei, E. & Haberlandt, U. (2014). Geostatistical merging of rain gauge and radar data for high temporal resolutions and various station density scenarios.

Journal of Hydrology, 508, 88-101.

Jewell, S. A. & Gaussiat, N. (2015). An assessment of kriging-based rain-gauge-radar merging techniques. Quarterly Journal of the Royal Meteorological Society.

Wang, L.-P., Ochoa-Rodríguez, S., Onof, C. & Willems, P. (2015). Singularity-sensitive gauge-based radar rainfall adjustment methods for urban hydrological applications. Hydrology and Earth System Sciences, 19 (9), 4001-4021.

Please also note the supplement to this comment:
http://www.hydrol-earth-syst-sci-discuss.net/hess-2016-113/hess-2016-113-AC4-supplement.pdf

―――――――――――――――

[Figure]

[Figure]

**Fig. 1.** Refers to comment above.

[Figure]

[Figure]

**Fig. 2.** Refers to comment above.

---

## Author Response (AR2)

**Anonymous Referee #1 (Report #2)**
Submitted on 01 Jul 2016

The authors deeply reorganized the manuscript improving its presentation and making it easier to follow. Nevertheless, most of the comments by the reviewers have not been addressed. According to the authors, this is due to technical problems, in fact the presented analysis has been carried out on operational runs that cannot be repeated due to insufficient resources.

The problem concerns the most scientifically interesting part of the manuscript, i.e. the assessment of the inclusion of the lightning data into radar and radar-gauge merged rainfall accumulation products. The authors are unable to find any significant improvement in the rainfall accumulation estimates. Such an outcome, if supported by robust analyses, represents a standalone result deserving publication. Nevertheless I am still convinced that the presented outcome is caused by a significant flaw in the data analysis that depends on the scales used for the comparison - please check my first review - and is therefore not supported by scientific evidence.

For these reasons I have to suggest the rejection of the manuscript from this special issue.

Nevertheless, given the interest of the topic, the very good operational system used and the efforts already devoted to the issue, I strongly invite the authors to take some time to improve the analyses and to resubmit an updated version of the manuscript.

Authors answer:
We have reevaluated our datasets from summer 2015. Here we found one mistake (in the code retrieving and calculating the dependent and independent stations). After correcting for this, we calculated the data for 2015 all over again. The outcome was that there are enough independent data for the period of 25 of intensive lightning days, which is now included into the verification result. We also understood, that as gauges are not used in the Rad_Accum and Rad_LDA_Accum methods, all gauges are independent as their verification references. The text and tables have been changed accordingly.

We have now put an extensive effort to follow the reveiwer's advice and included new results from a "scaled" dataset. For this, a new dataset with 25 intensive lightning cases from 2016 have been manually selected. We have scaled down the dataset, both by the geography and the time-intervals, to only include observations affected by lightning during these 25 days. We have analysed the outcome and even studied the distance dependancy to radar stations, in order to take into account other reveiwers suggestions. The new results from 2016 are included into this revised version of article and we can show an improvment using the LDA-method within LAPS accumulation process. See the new reorganized table 1 and related text in the result and discussion sections.

We want to point out that LDA is originally intended for complementing radars in the USA (where shadowing mountains create remarkable areas of missing data, and Ocean areas are included in analysis, without radar or gauge coverage). In this study we include LDA into an already existing operational analysis system, in order to complement the radar information and thereby improving the precipitation accumulation analysis but also other processes within the LAPS processes (like 3D clouds and moisture, which is of interest for LAPS-users but not a target in this specific publication). These are new developments in LAPS, using new methodologies, which might not be perfect from start but would need several years of further developments. We think there is value of publishing/sharing this work and by that, reach other workers in the same field, doing similar developments and possible help others to start from an advanced development point, using radar-lightning information.

One of the main result is that lightning is useful in areas of no radar data! If there is no radar, LAPS accumulation process has no way to continue, it need the reflectivity field. Therefore the lightning information can be crucial, it is there to complement the radar. There are areas (and even countries) which does not have radars, in souther latitudes the lightning is more frequent and here the LAPS-LDA model could be of even more importance, than in for example Finland.

Even though we here in Finland have a very good radar network, the results are improved.

**Referee #2: Dr. Hidde Leijnse, hidde.leijnse@knmi.nl (Report #3)**
Submitted on 08 Jul 2016

The readability of the paper has greatly improved by restructuring the paper. The main aim of the paper is now also much clearer. The authors have taken most reviewer comments seriously, and have implemented changes in the paper accordingly. However, the main issue that I still have with the paper is that the amount of data that is used for the analyses of the impact of using lightning data on the quality of the final precipitation estimates is not sufficient to draw any conclusions. Although I understand that it takes a large effort to produce a longer dataset, I still think that this is necessary in order for these analyses to have scientific value. If I understand it correctly, these analyses only require radar, gauge, and lightning data (and not NWP model data or satellite data), for which I know there are archives at FMI. Using such a dataset, this would also allow the selection of more independent rain gauges, and it would allow investigating whether there is an optimal integration time between 1 and 6 hours for the RandB method.

As two of the reviewers remarked that using linear scales for the scatter diagrams (and the ones with the contours), I would really like to urge the authors to use these linear scales. Especially because I found the graphs using linear axes that were provided with the response to the reviewers more instructive than the ones with logarithmic axes currently used in the paper.

So I am still of the opinion that the paper needs additional analyses of radar merged with lightning data before it is ready for publication.

Authors answer:
We have reevaluated our datasets from summer 2015. Here we found one mistake (in code retrieving and calculating the dependent and independent stations). After correcting for this, we calculated all the data for 2015 over again. The outcome was that there are enough independent data for the period of 25 of intensive lightning days, which is now included into the verification result. We also understood, that as gauges are not used in the Rad_Accum and Rad_LDA_Accum methods, all gauges are independent as their verification references. The text and tables have been changed accordingly.

We must again state that building up a new "experimental" LAPS-LDA system, to rerun periods with extensive datasets not being directly available (unless fetching form archive) would require a very extensive amount of work. In fact, also the NWP model data is required in order to run the LAPS accumulation process. The accumulation process module in LAPS is also producing other parameters, which relay on input data such as surface parameters and 3D-temperature (see http://laps.noaa.gov/software/README.html#ACCUM). Retreiving, unpacking, formatting and converting all involved datasets would require tools and systems to be combined and setup in an environment that does not yet exists, i.e. a routine that is outside LAPS operational routines. Please also remember that we are talking about massive data here (5 minutes of volume radar data, 3D NWP model data etc) and requirement of run-time demands on supercomputer facilites. This is simply not doable within in the limited amount of worktime and funding available.

Though, in order to follow the reviewer's suggestion to increase the dataset used for verification, we have spent much time to prepare and validate the new data output from summer 2016. Here we have followed the other reviewers suggestions on verifying "scaled" dataset and the distance dependeny from radar stations. With these new results, we are now confident that there are sufficient amount of data to draw the conclusion that lightning data and the LDA-mehtod does improve the accumulation estimates in LAPS. This is now included in the updated version of article, please see table 1 and the

related text in results and discussion sections. Note that we reorganized and merged the tables to make the readability and comparison easier.

Also, as suggested, we have now changed to use linear scales within the figures with contour plots (Figs. 4 and 7) and the scatter diagrams (Figs. 5 and 6).

I was reviewer # 3 for the initial submission of the paper.

I appreciate the improvement of the readability of this paper on a relevant topic. However I still think that the analysis in its current state does not enable to conclude on the interest or not of the use of LDA. The authors basically chose not to address all the serious concerns raised by me (and the other reviewers) with regards to the fact that the methodology suggested does not enable to reach a conclusion (either positive or negative that is not the point) with regards to the use of LDA. I understand that not everything is possible with operational data, but still think that more should be done before publication. See comments below with some suggestions of potential analysis with available (from my understanding) data. The innovative aspects of the paper should also be stressed more clearly.

Authors answer:
We appreciate the comments and suggestions by the reviewer.

We have reevaluated our datasets from summer 2015. Here we found one mistake (in code retrieving and calculating the dependent and independent stations). After correcting for this, we calculated all the data for 2015 over again. The outcome was that there are enough independent data for the period of 25 of intensive lightning days, which is now included into the verification result. We also understood, that as gauges are not used in the Rad_Accum and Rad_LDA_Accum methods, all gauges are independent as their verification references. The text and tables have been changed accordingly.

In order to reach a clear conclusion, whether LDA is useful or not, we have spent much time to prepare and validate the new data output from summer 2016. Here we have followed the reviewer's suggestion on verifying the distance dependeny from radar stations. With these new results, we are now confident that there are sufficient amount of data to draw the conclusion that lightning data and the LDA-mehtod does improve the accumulation estimates in LAPS. This is now included in the updated version of article, please see table 1 and the related text in results and discussion sections. Note that we reorganized and merged the tables to make the readability and comparison easier.

Detailed comments:

1) Introduction

- Some improvement, but not really a state of the art on assimilation of LDA for QPE. This would also help in understanding the novelties brought by the paper.
Authors answer:
We have reviewed all recent literature we could find. The problem with novel assimilation methods, as well as with other operational activities, seems to be that much of development happens in operational departments, where colleagues do not publish in journals that often. One of the motivations for this paper is actually to lower the wall between operational and scientific communities.

- l. 108-111 : this is a bit disappointing for a scientific paper... although the argument is partly valid, I still believe that more analysis can be done with the data available (see suggestions below).
Authors answer:
Setting up an LAPS-LDA system for rerunning experiments would require a very extensive amount of work. In fact, archived data from radars, NWP models, surface stations and lightning are required in

order to run the LAPS accumulation process. To retreive, unpack, format and convert all involved datasets would require tools and systems (to be combined and setup) in an environment that does not yet exists (i.e. a routine that is outside LAPS operational system and therefore not using operational input, on the contrary archived input). Please also remember that we here talk about massive data amounts (5 minutes of volume radar data, 3D NWP model data etc) and the requirement of run-time on supercomputer. This is simply not doable within in the limited amount of work-time and funding available.

Though, with much effort we have been able to follow the suggestion by the reviewer on distance dependancies, for which we used the most recent data as of summer 2016 (see below). Additionally we analysed the "scaled" datasets for this period (see below).

**2) Observation and instrumentation**

Please add a subsection on the event / period (summer / 25 days / 4 days) selection to clearly present them (moving the paragraph from the method section).
Authors answer:
We have now moved (and rewritten) the explanation of periods/events from section 3.5 to a new: Sect. 2.4.

**3) Methods**

- l.181 : "uses statistical methods", as already pointed out in the previous review, this formulation is not appropriate for a scientific paper, please be more specific.
Authors answer:
This has now been changed/removed.

- l. 227-228 : please clarify, how the quantiles are computed with regards to altitude (computed for each altitude). How is done the variable Quartile rad-Lig profiles. Since it is mentioned as a new method introduced by the authors, it should be more discussed, also in the results section.
Authors answer:
Information about calculation at altitudes is now included to the sentence: "For each of these 6 categories, the average reflectivity is calculated at each grid-point, for each level, and gives the Average Rad-Lig profiles (Fig. 3a) which is the baseline method."
More explanations on how the Variable Quartile profiles are calculated are now included to the text: "The specific percentiles used for the 6 categories are the 50-, 50-, 60-, 75-, 90- and 95% percentiles, respectively. The reason is to take into account the larger uncertainties in low categories (due larger spread and bias in the collected datasets) and on the other hand, rely on the higher percentiles for the higher categories (since these have less spread)."
We have now included some more text about Variable Quartile in the result section.

- l. 232-234 and figure 3.b : there seems to be somehow two regimes (at least one point -the one with the smallest number of strokes) seems isolated.
Authors answer:
Using a logarithmic scale and bins for lightning rate make the first category (the first point) contain mainly the weakest storms with one lightning stroke. Changing the bins might make the goodness of fit even better, however, we believe that R-square value of 0.95 is still relatively good.

- l. 238-241 : I still find it weird to chose simply maximum and was not convince by the answers of the

authors. It seems that this rule also applies when you are close from the radar where radar data is much more reliable. Why not testing other merging schemes that would for example take into account the distance to the radar or the attenuation. I guess that you have access to both radar mosaic and lightning maps separately (given that it is what is plotted in Fig. 8) ? This could add some novelties to the paper.

Authors answer:

We have now mentioned the uncertainty aspect in the article. We still believe that this is a valid scheme, to be used as a baseline, in our relatively dense radar network.

We are agree, there are many techniques/developments that could be tested, especially the radar-station dependancy has been discussed. Though, it would require a new study to find out the distance dependancy of radar and attenuation, which could be considered in upcoming versions of LAPS-LDA system.

In this paper we present the first results, using the first version of LAPS-LDA system, which we think bring novelties to the research, since these methods have not been used before. In a couple of years we hopefully have refined the techniques/models to perform even better. But again, we had to start from a baseline method and evolve from that. Therefore these are the results we have so far and which we think are important to share with others in same field of work.

- l. 244 : please remind the unit of Z (mm6 m-3)

Authors answer:

Yes, this is the correct unit and it is now put into the text.

- 3.3.1 : as already pointed out, it is weird to have a 3.3.1 and no 3.3.2...

Authors answer:

We have now change section 3.3.1 to be section 3.4.

- 3.4 : what is the interest of presenting results for periods without lightning ? l.312 : remind here why only 7 independent stations are used (as explained in the answer to reviewers)

Authors answer:

It is a standard procedure when introducing new components in the assimilation schema to make sure that these do not have a harmful impact in cases of missing data. Hence, we have investigated the impact by LDA-method over several periods, to prove robustness and impact of the new developments in an operational environement. It is important to verify that the system works for all different weather situations. We have reorganzed the tables into only one table (thereby including new results and improving the comparabiltiy of data) and in order to follow the reveiwer's suggestion, we have now removed the information on the long summer period of 2015 from the table content. Instead this is only mentioned in the result section's text, but there is still the plot in figure 4.

Since the verification was performed during operational runs, the independent stations had to be set beforehand (i.e. excluded from the assimilation). Because of this we could not set more stations aside, without risking the quality of the end product. The seven independent stations were selected subjectively from different physiographical areas such as coastline, inland, lake district, and proximity to each other. On average, within a radius of 50 km from the independent station point, there are 11 dependant stations and the average distance to the nearest dependant station is 9.8 km. Please also see the notification at top of this document, which relate to the independent stations.

4) Results

- l. 335-345 : it is frustrating that no lightning occurred over the 7 independent stations during the 25 selected events... Given that it seems some lightning was recorded for some events (since an

improvement is noticed), why not selecting a sub-set of event for which lightning was observed over the independent stations and discuss the results for them.

Authors answer:
Please see the notification at top of this document, which relate to the independent stations.

We went over the data from 2015 and discovered one mistake in the programs that we used. Now we are able to include independent data for the 25 lightning intensive days from 2015 into the result (see Table 1). Also, the Rad_Accum and Rad_LDA_Accum are data that are independent dataset, since gauges are not involved when estimating the accumulation with these methods.

In the data from 2016, 25 lightning days, we have made a "scaled" subset of data. We have manually scaled down the dataset, both by the geography and the time-intervals, to only include observations affected by lightning during these 25 days.  See the new reorganized table 1 and related text in the result and discussion sections.

- l. 359-360 : comments on Fig 6 : "The 3'rd Quartile gives the largest overestimate, over the whole accumulation scale", it seems rather obvious no ? An explanation on the variable Quartile profile is missing.

Authors answer:
Yes, it might be obvious (for most of us) but should still be mentioned in the results.

We have added the following text related to Variable Quartile: "*The Variable Quartile gives the overall best result, with improved estimates for high accumulation values and only slight overestimation at low values.*"

- Section 4.2 : please give more explanations and also comments on the curves of Fig. 7.

Authors answer:
The curves in Fig. 7 can be seen as isolines of density of points in a scatter plot. We have found this approach more useful than scatter plots when the points would overlap each other, and the approach was also recommended by reviewers of our earlier paper.

We have added more text into section 4.2 and figure caption.

5) Discussion and conclusion (by the way add a number for this title section)

Authors answer:
We have added the Section number (5).

- l.374 : "at least leave them as unaffected as possible", somehow weird formulation as a goal...

Authors answer:
This sentence has been changed.

- l. 375-378 : this is indeed an interesting point but not demonstrated.... it could be done with the available data (i.e. not re-running past events) by considering scores for stations located only far (testing different distances) from the radars (ex. : stations with distance to the radar equal to 0-50 km, 50-100km, 100-150km...).

Authors answer:
We have now made a big effort to follow the reveiwer's suggestion on analyzing the distance dependancy of radar station location and gauge stations, to see the effect on accumulation scores. For this we used the newest dataset from summer 2016, which uses the Variable Quartile profiles. The results can be seen in Table 1 and related text is included into result and discussion sections. These new results shows improved scores (because of the use of Variable Quartile method) and we can conclude

the usefullness of assimilating lightning data in the LAPS accumulaiton process.

- l. 404-406 : "3rd quartile gives higher impact....", I may be confused, but it is somehow surprising the median does not work better on average (was the estimation correct). Please discuss this point more.
Authors answer:
We are not sure if the reviewer here means "mean" instead of "median"? In the article we have used Average- and Quartile profiles. We have also made some unpublished testing with Median, which then lead us to test different Quartile approaches, showing that the Variable Quartile method outperformed both Average- and Median methods.
When collecting the corresponding radar profiles from the different bins (as in Fig. 3a), the spread/bias are relatively larger for the lowest bins (i.e. with few lightning strikes) than for the higher bins. The spread comes from different factors, such as the mismatch between instruments, lightning might not occur right above the precipitation and there is of course a fluctuation of amount of lightning for each storm event. There are many uncertainties involved and by building the statistics for a long time-period, some of the uncertainties are minimized.
The lowest bins (less lightning) are related to a very widespread range of precipitaion amounts (here we have the biggest uncertainties and with longer statistics this spread becomes even larger). Therefore, it is "safer" to use the average (e.g. 50% percentile) value to calculate the profiles for the lowest 2 bins. The higher bins (intense lightning) mainly occurs during intense rainfall and here is less spread in the data, so using a 90-95% percentile for these cases are realistic and our testruns have proven that.
By using the Variable Quartile approach we try to include these uncertainties, i.e. a sliding scale of changing percentiles is used (ranging from 50-95%), where average is used for lowest 2 bins and 95% percentile for the highest bin.

[revised manuscript text omitted]
 stations show some improvement (Table 2). The correlation (i.e.verification give decreased RMSE and increased CORR) is higher values for the Rad_LDA_Accum independent data (compared to Rad_Accum), and even though the RMSE is higher, the STDEV has been improved. The overall impact for . Also, Rad_LDA_RandB get smaller errors than Rad_RandB (see STDEV and RMSE in Table 1; most upper right panel). For the dependent stations is neutral (Table 2 and Fig. 4; compare Rad_Accum and Rad_LDA_Accum). For both the 25 days and whole summer period the Rad_RandB and Rad_LDA_RandB give the best results, with similar scores.
[revised manuscript text omitted]

| Page 24: [1] Formatted | E. Gregow | 06/09/2016 13:57:00 |

Font color: Auto

| Page 24: [1] Formatted | E. Gregow | 06/09/2016 13:57:00 |

Font color: Auto

| Page 24: [1] Formatted | E. Gregow | 06/09/2016 13:57:00 |

Font color: Auto

| Page 24: [1] Formatted | E. Gregow | 06/09/2016 13:57:00 |

Font color: Auto

| Page 24: [1] Formatted | E. Gregow | 06/09/2016 13:57:00 |

Font color: Auto

| Page 24: [1] Formatted | E. Gregow | 06/09/2016 13:57:00 |

Font color: Auto

| Page 24: [1] Formatted | E. Gregow | 06/09/2016 13:57:00 |

Font color: Auto

| Page 24: [1] Formatted | E. Gregow | 06/09/2016 13:57:00 |

Font color: Auto

| Page 24: [1] Formatted | E. Gregow | 06/09/2016 13:57:00 |

Font color: Auto

| Page 24: [2] Formatted | E. Gregow | 06/09/2016 13:57:00 |

Font: 9 pt, Bold, Font color: Auto, German (Germany)

| Page 24: [3] Formatted | E. Gregow | 06/09/2016 13:57:00 |

Table Contents

| Page 24: [4] Formatted | E. Gregow | 06/09/2016 13:57:00 |

Font: 8 pt, Font color: Auto, German (Germany)

| Page 24: [5] Formatted | E. Gregow | 06/09/2016 13:57:00 |

Table Contents

| Page 24: [6] Formatted | E. Gregow | 06/09/2016 13:57:00 |

Font: 8 pt, Font color: Auto, German (Germany)

| Page 24: [7] Formatted | E. Gregow | 06/09/2016 13:57:00 |

Font: 8 pt, Font color: Auto, German (Germany)

| Page 24: [7] Formatted | E. Gregow | 06/09/2016 13:57:00 |

Font: 8 pt, Font color: Auto, German (Germany)

| Page 24: [8] Inserted Cells | E. Gregow | 06/09/2016 13:57:00 |

Inserted Cells

| Page 24: [9] Formatted | E. Gregow | 06/09/2016 13:57:00 |

Font: 8 pt, Font color: Auto, German (Germany)

| Page 24: [10] Formatted | E. Gregow | 06/09/2016 13:57:00 |

Table Contents

| Page 24: [11] Formatted | E. Gregow | 06/09/2016 13:57:00 |

Font: 8 pt, Font color: Auto, German (Germany)

| Page 24: [11] Formatted | E. Gregow | 06/09/2016 13:57:00 |

Font: 8 pt, Font color: Auto, German (Germany)

| Page 24: [12] Inserted Cells | E. Gregow | 06/09/2016 13:57:00 |

Inserted Cells

| Page 24: [13] Inserted Cells | E. Gregow | 06/09/2016 13:57:00 |

Inserted Cells

| Page 24: [14] Inserted Cells | E. Gregow | 06/09/2016 13:57:00 |

Inserted Cells

| Page 24: [15] Inserted Cells | E. Gregow | 06/09/2016 13:57:00 |

Inserted Cells

| Page 24: [16] Formatted | E. Gregow | 06/09/2016 13:57:00 |

Font: 8 pt, Font color: Auto, German (Germany)

| Page 24: [17] Formatted | E. Gregow | 06/09/2016 13:57:00 |

Table Contents

| Page 24: [18] Formatted | E. Gregow | 06/09/2016 13:57:00 |

Font: 8 pt, Font color: Auto, German (Germany)

| Page 24: [19] Inserted Cells | E. Gregow | 06/09/2016 13:57:00 |

Inserted Cells

| Page 24: [20] Inserted Cells | E. Gregow | 06/09/2016 13:57:00 |

Inserted Cells

| Page 24: [21] Formatted | E. Gregow | 06/09/2016 13:57:00 |

Font: 9 pt, Font color: Auto, German (Germany)

| Page 24: [22] Formatted | E. Gregow | 06/09/2016 13:57:00 |

Table Contents

| Page 24: [23] Formatted | E. Gregow | 06/09/2016 13:57:00 |

Font: 9 pt, Font color: Auto, German (Germany)

| Page 24: [24] Formatted | E. Gregow | 06/09/2016 13:57:00 |

Font: 9 pt, Font color: Auto, German (Germany)

| Page 24: [25] Formatted | E. Gregow | 06/09/2016 13:57:00 |

Font: 9 pt, Font color: Auto, German (Germany)

| Page 24: [26] Formatted | E. Gregow | 06/09/2016 13:57:00 |

Font: 9 pt, Font color: Auto, German (Germany)

| Page 24: [27] Formatted | E. Gregow | 06/09/2016 13:57:00 |

Font: 9 pt, Font color: Auto

| Page 24: [28] Inserted Cells | E. Gregow | 06/09/2016 13:57:00 |

Inserted Cells

| Page 24: [29] Formatted | E. Gregow | 06/09/2016 13:57:00 |

Font: 9 pt, Font color: Auto, German (Germany)

| Page 24: [30] Formatted | E. Gregow | 06/09/2016 13:57:00 |

Table Contents

| Page 24: [31] Formatted | E. Gregow | 06/09/2016 13:57:00 |

Font: 9 pt, Font color: Auto, German (Germany)

| Page 24: [32] Formatted | E. Gregow | 06/09/2016 13:57:00 |

Font: 9 pt, Font color: Auto, German (Germany)

| Page 24: [33] Formatted | E. Gregow | 06/09/2016 13:57:00 |

Font: 9 pt, Font color: Auto, German (Germany)

| Page 24: [34] Inserted Cells | E. Gregow | 06/09/2016 13:57:00 |

Inserted Cells

| Page 24: [35] Inserted Cells | E. Gregow | 06/09/2016 13:57:00 |

Inserted Cells

| Page 24: [36] Formatted | E. Gregow | 06/09/2016 13:57:00 |

Table Contents

| Page 24: [37] Formatted | E. Gregow | 06/09/2016 13:57:00 |

Font: 9 pt, Font color: Auto, German (Germany)

| Page 24: [37] Formatted | E. Gregow | 06/09/2016 13:57:00 |

Font: 9 pt, Font color: Auto, German (Germany)

| Page 24: [38] Formatted | E. Gregow | 06/09/2016 13:57:00 |

Font: 9 pt, Font color: Auto, German (Germany)

| Page 24: [38] Formatted | E. Gregow | 06/09/2016 13:57:00 |

Font: 9 pt, Font color: Auto, German (Germany)

| Page 24: [39] Formatted | E. Gregow | 06/09/2016 13:57:00 |

Font: 9 pt, Font color: Auto, German (Germany)

| Page 24: [40] Formatted | E. Gregow | 06/09/2016 13:57:00 |

Table Contents

| Page 24: [41] Formatted | E. Gregow | 06/09/2016 13:57:00 |

Font: 9 pt, Font color: Auto, German (Germany)

| Page 24: [41] Formatted | E. Gregow | 06/09/2016 13:57:00 |

Font: 9 pt, Font color: Auto, German (Germany)

| Page 24: [42] Formatted | E. Gregow | 06/09/2016 13:57:00 |

Font: 9 pt, Font color: Auto, German (Germany)

| Page 24: [43] Inserted Cells | E. Gregow | 06/09/2016 13:57:00 |

Inserted Cells

| Page 24: [44] Formatted | E. Gregow | 06/09/2016 13:57:00 |

Table Contents

| Page 24: [45] Formatted | E. Gregow | 06/09/2016 13:57:00 |

Font: 9 pt, Font color: Auto, German (Germany)

| Page 24: [45] Formatted | E. Gregow | 06/09/2016 13:57:00 |

Font: 9 pt, Font color: Auto, German (Germany)

| Page 24: [46] Formatted | E. Gregow | 06/09/2016 13:57:00 |

Font: 9 pt, Font color: Auto, German (Germany)

| Page 24: [47] Formatted | E. Gregow | 06/09/2016 13:57:00 |

Font: 9 pt, Font color: Auto, German (Germany)

| Page 24: [48] Formatted | E. Gregow | 06/09/2016 13:57:00 |

Table Contents

| Page 24: [49] Formatted | E. Gregow | 06/09/2016 13:57:00 |

Font: 9 pt, Font color: Auto, German (Germany)

| Page 24: [50] Formatted | E. Gregow | 06/09/2016 13:57:00 |

Font: 9 pt, Font color: Auto, German (Germany)

| Page 24: [51] Formatted | E. Gregow | 06/09/2016 13:57:00 |

Table Contents

| Page 24: [52] Formatted | E. Gregow | 06/09/2016 13:57:00 |

Font: 9 pt, Font color: Auto, German (Germany)

| Page 24: [52] Formatted | E. Gregow | 06/09/2016 13:57:00 |

Font: 9 pt, Font color: Auto, German (Germany)

| Page 24: [53] Formatted | E. Gregow | 06/09/2016 13:57:00 |

Font: 9 pt, Font color: Auto

| Page 24: [54] Inserted Cells | E. Gregow | 06/09/2016 13:57:00 |

Inserted Cells

| Page 24: [55] Inserted Cells | E. Gregow | 06/09/2016 13:57:00 |

Inserted Cells

| Page 24: [56] Formatted | E. Gregow | 06/09/2016 13:57:00 |

Table Contents

| Page 24: [57] Formatted | E. Gregow | 06/09/2016 13:57:00 |

Font: 9 pt, Bold, Font color: Auto, German (Germany)

| Page 24: [57] Formatted | E. Gregow | 06/09/2016 13:57:00 |

Font: 9 pt, Bold, Font color: Auto, German (Germany)

| Page 24: [58] Inserted Cells | E. Gregow | 06/09/2016 13:57:00 |

Inserted Cells

| Page 24: [59] Inserted Cells | E. Gregow | 06/09/2016 13:57:00 |

Inserted Cells

| Page 24: [60] Formatted | E. Gregow | 06/09/2016 13:57:00 |

Table Contents

| Page 24: [61] Formatted | E. Gregow | 06/09/2016 13:57:00 |

Table Contents

| Page 24: [62] Formatted | E. Gregow | 06/09/2016 13:57:00 |

Font: 9 pt, Font color: Auto, German (Germany)

| Page 24: [63] Formatted | E. Gregow | 06/09/2016 13:57:00 |

Font: 9 pt, Font color: Auto, German (Germany)

| Page 24: [64] Formatted | E. Gregow | 06/09/2016 13:57:00 |

Table Contents

| Page 24: [65] Formatted | E. Gregow | 06/09/2016 13:57:00 |

Font: 9 pt, Font color: Auto, German (Germany)

| Page 24: [65] Formatted | E. Gregow | 06/09/2016 13:57:00 |

Font: 9 pt, Font color: Auto, German (Germany)

| Page 24: [66] Formatted | E. Gregow | 06/09/2016 13:57:00 |

Font: 9 pt, Font color: Auto, German (Germany)

| Page 24: [66] Formatted | E. Gregow | 06/09/2016 13:57:00 |

Font: 9 pt, Font color: Auto, German (Germany)

| Page 24: [67] Inserted Cells | E. Gregow | 06/09/2016 13:57:00 |

Inserted Cells

| Page 24: [68] Inserted Cells | E. Gregow | 06/09/2016 13:57:00 |

Inserted Cells

| Page 24: [69] Inserted Cells | E. Gregow | 06/09/2016 13:57:00 |

Inserted Cells

Inserted Cells

Inserted Cells

Inserted Cells

Inserted Cells

---

## Author Response (AR3)

**Reviewer/Report #1**
Submitted on 20 Oct 2016
(Anonymous Referee #3)

This is the third time that I review this paper. I appreciate the additional work carried out by the authors and corresponding improvement to the paper. Although not all comments of the reviewer have been taken into account mainly because of practical difficulties, I think that the paper can now be published. Nevertheless I invite the authors to continue their work on this topic and publish results with more extensive data sets in the future.

**Authors: We want to thank the reviewer for the effort of reviewing this article. This has improved the readability and quality of the paper, which we are very grateful for.**
* * *
**Review/Report #2**
Submitted on 30 Oct 2016
(Anonymous Referee #1)

I am the anonymous referee #1 of the previous revision and I'll mainly focus on the response to my previous revision. The authors now put extensive efforts in improving the analysis and providing more interesting results, nevertheless, in my opinion, the manuscript still needs to be improved before being published, mainly because the results are not clearly presented. See my major comments below:

1. First, I'd like to answer the last of the author's response: *"We think there is value of publishing/sharing this work and by that, reach other workers in the same field, doing similar developments and possible help others to start from an advanced development point, using radar-lightning information. One of the main result is that lightning is useful in areas of no radar data! If there is no radar, LAPS accumulation process has no way to continue, it need the reflectivity field. Therefore the lightning information can be crucial, it is there to complement the radar. There are areas (and even countries) which does not have radars, in souther latitudes the lightning is more frequent and here the LAPS-LDA model could be of even more importance, than in for example Finland".*
This is very interesting to know and motivates part of the study, but the reader does not necessarily know it. I highly recommend to include it in the introduction (motivation for this study), to highlight it in the proper sections of the discussion, and to recall it in the conclusions (see also point 2). For example, lines 205-211 could be moved to the introduction for this purpose.

Authors: We have now pointed out these motivations in both introduction and in discussion/conclusion sections.

2. The "verification periods" are not presented and used clearly. From what I understood from section 2.4, the authors use 5 different verification periods: (a) summer 2015; (b) summer 2016; (c) 25 days during summer 2015; (d) 25 days with "scaled" data during summer 2016 and (e) 4 days with high lightning occurrence during summer 2014. I suggest to organize the periods more clearly, for example numbering them (like I did here).

Authors: We have rephrased section 2.4 and followed the suggestions by the reviewer.

- How is the data in (d) scaled? "Manually" is not enough to understand, did the authors select a maximum distance from the lightning? What about time?

Authors: This is now clarified and text added in section 2.4. *"(defined as stations with maximum distance of 30 km to the lightning position and within the 1 hour accumulation time-interval, hereafter called scaled dataset)"*.

- When is (d) used? In line 193 the authors introduce (d): "hereafter called scaled dataset", but the word "scaled" is then used again only once, in line 395. I believe that (d) is used also in Table 1, am I right?

Authors: This is correct. We have added clarifications to when this dataset is being used in section 4.1 and added this also to table 1.

- In the presentation of the results (section 4.1) the use of these periods is not clear: what periods are presented in Table 1? The caption says that the results are presented for (c) and (d) but the text is not clear on this.

Authors: Section 4.1 has now been updated with explanations of which verification dataset are being used. Additionally, all tables and figures are updated with same information.

- Lines 362-367 say that the verification on (a) and (b) gives no significant impact, so the results are not presented. SO, why are these verification periods included? I imagine the reason is what discussed in the point 1 above, so I highly recommend to discuss this here and to present Fig. 4 with this purpose.

Authors: We are making a gradual down-scaling in time. Starting from long-term verification to investigate the effect from LDA and this is also where we make sure the system works in every situation (sanity-check). After that we narrow down the study to locate the details of interest and the explicit impact of lightning data. We have changed the text in this paragraph, to make this more clear: *"The verification for the entire summer of 2015, i.e. using verification dataset a) including days with no thunderstorms, assures that introducing lightning data has no significant impact in the overall performance of the system. The impact by using LDA-mehtod for estimating the precipitation accumulation is neutral, for this long verification period (shown in Fig. 4, where the data are from dependent stations). Same result is seen in the scores of RMSE, STDEV and CORR values (not included here). Since the data have been much influenced by weather situations not relating to lightning, the focus will be on the subsets, i.e. datasets c) and d), the 25 days periods of intense lightning days of both 2015 and 2016, respectively."*

- Fig 4: is the figure from the verification period (a)? or from (c)? Is this the figure the authors refer to in lines 362-367? Perhaps some statistic value could provide the same message more clearly.

Authors: Yes, figure 4 is related to verification dataset a), summer 2015. This is now clarified both in the text and the figure 4 caption. Also, we have now added the regression line equations in figure 4, which makes it possible to compare them in a quantative way.

- What verification periods are used for the integration period length analysis (sec 4.2, Fig. 7)? This information is not provided to the reader.

Authors: Thank you, this was missed. Here the verification dataset a) was used. This has now been added, both to section 4.2 and figure 7.

3. Results are presented not clearly. They need to be reorganized and their presentation needs to be improved.

- Line 369: what does "shows a biased result"?

Authors: We have changed to use the wording "inconsistent". At this line we rephrased the sentence including "biased" to read *"The subset of 2015, using the Average method, gave inconsistent results and no unambiguous conclusions could be drawn (Table 1, left column)"*.

- Why are some results presented via table and statistical metrics and other via figures? This is not clear and does not allow to compare between them. Results from the figures are qualitative, since no information on the regression lines, on the bias or other statistic is provided.

Authors: We wanted to find a good combination of different ways of presenting the material, in a reader-friendly and interesting way. Instead of tabulated values we wanted to present the whole summer period dataset in a graph and let the reader get an understanding of how the data looks like in a plotted graph, instead of table. We have now added the regression line equations in figures 4-7, in order to compare the results in graphs in a quantitative way.

- Line 385: "most of the scores are improved", the improvements observed from the tables are actually minor (the highest being RMSE for the dependent dataset of 2016 decreased by 6.3%), some words on the statistical significance of the improvements or on the idea expressed in the point 1 above could be useful.

Authors: The sentence have been rephrased as ""*Even if the improved scores are relatively small (largest reduction in RMSE being 6.3%) the LDA-method show a consistent correction of the results"*. Improving the accumulation estimates is not a simple task and even the slightest improvement is important. Especially if it related to heavy precipitating convective situations with lightning, then a 6% improvement can be critical. We believe that this is a notable improvement, which show the benefit of using this method. Also, the work is on-going and these scores can potentially become better with future developments.

- Line 394-396: these lines are more useful at the beginning of this paragraph, in order to introduce the results.

Authors: We have moved the sentence to the beginning of same paragraph.

- Table 1 presents different verification periods (period (c) and (d) of point 2 above, if I understood correctly) and different schemes, over different verification datasets. This makes difficult to understand if the differences in performance are due to the different method, to the dataset or to the verification period.

Authors: We have now clarified the different verification periods used in the table text. We have to use the two different verification periods and datasets (i.e. 2015 and 2016), since we are not able to rerun longer periods (as explained before). The results show/indicate that the Variable Quartile scheme (results for 2016) performs better than the Average scheme (2015). With this table the reader is able to compare the results side-by-side, which makes it simple to draw conclusions of the performance/scores of different LDA-methods (i.e. different schemes) and distance dependencies. This table was constructed because of other reviewers comments (in previous revision round). We believe that this table is the best way to present the material in a comparable way, better than for example to present the results in separate tables. Therefore, we would prefer to keep the table as it is (with the clarification of which verification datasets being used).

Minor comments:

- Abstract: there is no need of introducing the acronyms in the abstract, if they are not used therein.

Authors: Done

- Lines 49-54: not clear

Authors: These sentences have been clarified.

- Line 60: the acronym is not used anymore so there is no need to introduce it

Authors: The accronym is removed.

- Lines 71-75 would fit better in the methods

Authors: We have moved these two sentences to the end of section 3.2.

- Lines 105-119: this part is a bit verbose

Authors: Part of this text was added due to other reviewers comments, in previous revision round. We have now rewritten the paragraph and made it more concise: *"Gregow et al. (2013) have demonstrated the benefit of assimilating different data sources (radars and gauges) in precipitation estimation. The largest uncertainties were observed during heavy convective rainfall. These are the situations when lightnings occur. The accumulation process is based on radar reflectivity field, where gauges corrects the initial field, e.g. if there is no reflectivity field there is no accumulation (gauges are not used alone). To improve the spatially accurate real-time precipitation analysis new methods are adopted by fusion of weather radar, lightning observations and rain gauge information in novel ways. This leads to better possibilities in estimating convective rainfall events (i.e. > 5 mm/h) and the accumulated precipitation, for the benefit of hydropower management and other related application areas. The work reported here has been performed using the Local Analysis and Prediction System (LAPS), which is used operationally in the weather service of Finnish Meteorological Institute (FMI). Testing new*

*approaches in an operational system has its challenges. For example, it is not possible to exclude a large amount of independent reference stations. Also the possibilities to rerun cases with different settings have been limited. The major benefit of working in an operational environment is that we can be sure that we only use data and methods which are operationally available and feasible."*

- Line 112: I'd use "convective" rather than "extreme"

Authors: This has been changed.

- Lines 147-153: the names of the radars are not reported in the map; I suggest either using the same symbol as the maps or more generic words

Authors: We have changed to use the same station names as in figure 1b, also naming in figure caption is now changed.

- Line 164, 181, 194: LDA, LAPS, FMI-LAPS LDA have not been introduced yet

Authors: This is now corrected.

- Line 180-182: move to section 3.2

Authors: The sentence has been moved to section 3.2, first paragraph.

- Line 260-262: please provide more details

Authors: We have provided more indeepth explantion of the smoothnig of Rad-Lig profiles: *"The profiles from the two categories with largest amount of strokes have the least data, because they are the rarest categories. All datasets suffer from missing data at some height levels, but these two categories are more sensitive, due to the overall small data amounts. This can sometimes create artificial peaks of too low reflectivity values. This was especially seen at high altitudes, which can partly be explained by the radar measurement geometry. Therefore these two reflectivity profiles have been manually smoothened to have the same shape as the other profiles."*

- Lines 285-286: move to introduction

Authors: The sentence has been moved to the Introduction section.

- Line 291: "accumulating" rather than "by summing up over the"

Authors: This has been changed.

- Lines 308-309: gauges are not mentioned here but, if I understood correctly, represent a key aspect of the Rand-B method

Authors: Correct, "gauges" has been added to the sentence.

- Lines 337-339: move before in the paragraph

Authors: We have moved this sentence.

- Line 342: before in the text the author said that no beam blockage affects the area

Authors: We are sorry for the proofreading error, the "beam blocking" is now corrected to "beam broadening" with an explanation that "(*1 degree beam is 5 km wide at distance of 250 km)*".

- Fig 5 and 6: use the same words for corresponding features in the figures

Authors: This has been corrected in figure captions 5 and 6.

[revised manuscript text omitted]